# Multi-model ensemble projection of global dust cycle by the end of 21st century using CMIP6 data

**Yuan Zhao[1], Xu Yue[1], Yang Cao[2], Jun Zhu[1], Chenguang Tian[1], Hao Zhou[2], Yuwen Chen[1], Yihan Hu[1], Weijie Fu[1] and Xu Zhao[1]**

[1] Jiangsu Key Laboratory of Atmospheric Environment Monitoring and Pollution Control, Collaborative Innovation Center of Atmospheric Environment and Equipment Technology, School of Environmental Science and Engineering, Nanjing University of Information Science & Technology (NUIST), Nanjing, 210044, China

[2] Climate Change Research Center, Institute of Atmospheric Physics, Chinese Academy of Sciences, Beijing, 100029, China

Corresponding author: Xu Yue (Email: yuexu@nuist.edu.cn)

**Abstract**

As a natural aerosol with the largest emissions on land, dust has important impacts
on atmospheric environment and climate systems. Both the emissions and transport of
dust aerosols are tightly connected to meteorological conditions and as a result are
confronted with strong modulations by the changing climate. Here, we project the
changes of global dust emissions and loading by the end of the 21$^{st}$ century using an
ensemble of model outputs from the Coupled Model Intercomparison Project version 6
(CMIP6) under four Shared Socioeconomic Pathways (SSPs). Based on the validations
against site-level observations, we select 9 out of 14 models and estimate an ensemble
global dust emission of 2566±1996 Tg a$^{-1}$ (1Tg = 10$^{12}$g) at present day, in which 68%
is dry deposited and 31% is wet deposited. Compared to 2005-2014, global dust
emissions show varied responses with a reduction of -5.6±503 Tg a$^{-1}$ under the SSP3-
7.0 scenario but increased emissions up to 60.7±542 Tg a$^{-1}$ under the SSP5-8.5 scenario
at 2090-2099. For all scenarios, the most significant increase of dust emissions appears
in North Africa (0.6%-5.6%) due to the combined effects of reduced precipitation but
strengthened surface wind. In contrast, all scenarios show decreased emissions in
Taklimakan and Gobi Deserts (-0.8% to -11.9%) because of the increased precipitation
but decreased wind speed regionally. The dust loading shows uniform increases over
North Africa (1.6%-13.5%) and the downwind Atlantic following the increased
emissions, but decreases over East Asia (-1.3% to -10.5%) and the downwind Pacific
partly due to enhanced local precipitation that promotes wet deposition. In total, global
dust loading will increase by 2.0%-12.5% at the end of the 21$^{st}$ century under different
climate scenarios, suggesting a likely strengthened radiative and climatic perturbations
by dust aerosols in a warmer climate.

**Keywords:** CMIP6, dust emissions, concentrations, climate change, ensemble
projection


## 1 Introduction

Dust aerosol is one of the major air pollutants with strong climatic and environmental effects. Suspended dust aerosols can absorb and scatter solar radiation, and act as condensation nuclei so as to change the cloud optical properties (Tegen et al., 2004; Penner et al., 2006; Forster et al., 2008). Dust deposition can change the albedo of snow and ice and transport mineral elements to the ocean (Jickells et al., 2005; Mahowald et al., 2005; Wittmann et al., 2017). Furthermore, strong dust storms present as a serious threat to human society by reducing road visibility that influences traffic safety (Middleton, 2017), carrying bacteria and viruses that affects public health (Goudie, 2014), and reducing crop yields that endangers the food supply (Stefanski and Sivakumar, 2009). In light of the great impacts of dust on climate and environment, it is of significant importance to study the spatiotemporal characteristics and future changes of global dust aerosols.

The dust cycle consists of three major processes including emission, transport, and deposition (Schepanski, 2018), which are mainly related to meteorological conditions, such as precipitation, humidity, surface wind speed, and turbulent mixing (Liu et al., 2004; Shao et al., 2011; Csavina et al., 2014). Low humidity and/or strong surface wind are in favor of dust emissions (Csavina et al., 2014). Atmospheric humidity has a tight coupling effect with soil moisture, which in part controls the threshold of friction velocity and dust emission intensity (Munkhtsetseg et al., 2016). Strong winds and the associated pressure systems promote the momentum of surface layer and consequently increase dust mobilizations (Li et al., 2022). The transport of dust aerosols is related to atmospheric circulation and turbulent mixing, which determine the horizontal and vertical distribution of dust aerosol particles, respectively (Zhang et al., 2014; Fernandes et al., 2020). The deposition process includes dry and wet settlement, in which the dry deposition is an effective way to remove large particles while wet deposition dominates the removal of fine particles (Breuning-Madsen and Awadzi, 2005; Yue et al., 2009). Therefore, the spatiotemporal variations of dust aerosols are closely related to meteorological factors.

Climate change exerts significant impacts on the global dust cycle. A study using

the RegCM3 model showed that dust emissions and the column burden would increase
respectively by 2% and 14% in eastern Asia at 2091-2100 relative to 1991-2000 (Zhang
et al., 2016). In contrast, the earlier study projected the reductions of dust emissions by
26% using the ECHAM4-OPYC model and 19% using the HADCM3 model in the
same region by the midcentury (Tegen et al., 2004). Compared to these studies based
on 1-2 models, the ensemble projections using multiple models from the Climate Model
Inter-comparison Project (CMIP) showed great potentials of indicating the uncertainties
in the estimate of global dust cycle. Wu et al. (2020) evaluated 15 dust models in CMIP
phase 5 (CMIP5) and found that the uncertainty was relatively small for the dust belt
extending from North Africa to East Asia, but the uncertainties in other regions such as
Australia and North America were large. Based on the multi-model ensemble from
CMIP5 data, Pu and Ginoux (2018) estimated an increase of dust optical depth in
central Arabian Peninsula and a decrease over northern China in the late half of the 21$^{st}$
century under a strong warming scenario. Zong et al. (2021) also projected that dust
emissions would decrease in East Asia by the end of 21$^{st}$ century under the same climate
scenario. However, the different features of future global dust cycles and the related
drivers under varied climate scenarios remain unclear.

95       The recent phase 6 of CMIP (CMIP6) includes more complete dust variables (e.g.,

emissions, depositions, concentrations, and optical depth) from climate models. The
ensemble of CMIP6 simulations has been used to depict historical changes in dust cycle
and explore the possible climatic drivers (Le and Bae, 2022; Li and Wang, 2022).
However, this valuable dataset has rarely been used for the future projections on the
global scale. Compared to CMIP5 models, more dust emission schemes are coupled
with dynamic vegetation in CMIP6 models to optimize land surface emission processes
(Zhao et al., 2022). Such improvement may also amplify the uncertainties of dust
simulations, because the predicted vegetation change may be inconsistent with the
observed tendencies (Wu et al., 2020). As a result, it is important to validate the
simulated present-day dust cycle before the application of different models in the future
projection (Aryal and Evans, 2021). In this study, we project the future changes in
global dust cycles by the end of 21$^{st}$ century under four different climate scenarios based

on the multi-model ensemble mean from CMIP6 models. We select a total of 14 climate models providing dust emissions, depositions, and concentrations for all the four scenarios and validate the simulated near-surface dust concentrations and aerosol optical depth (AOD) with site-level measurements. The models with reasonable performance are selected to project future changes in dust emissions and loadings by the years 2090-2099 relative to the present day (2005-2014). The changes in associated meteorological conditions are further explored to identify the main causes of the changes in the global dust cycle.

**2 Methods and data**

2.1 Model data

We select all available CMIP6 models (last access: April 20$^{th}$, 2023) providing complete variables of dust cycle (emission, dry/wet deposition, and concentration) and the associated meteorology (surface wind, relative humidity, precipitation) for both present day and four future scenarios under the Shared Socioeconomic Pathways (SSPs) of SSP1-2.6, SSP2-4.5, SSP3-7.0, and SSP5-8.5, which represent future climate with the low to high anthropogenic radiative forcings. A total of 14 models with different spatial resolutions are selected (Table 1). Different models may have varied numbers of ensemble runs for dust cycle variables (Table S1). We use all available runs with different variants and labels from each of climate models, resulting in a total of 416 runs for every dust variable (120 for history and 296 for four future scenarios) and 770 runs for every meteorological variable (212 for history and 558 for four future scenarios). In addition, we collect both dust optical depth (DOD) and AOD at the historical periods from these models (Table S1). To facilitate the model validation and inter-comparison, we interpolate all model data with different spatial resolution to the same of 1°×1°. For each model, we average all the ensemble runs under one climatic scenario to minimize the uncertainties due to initial conditions. As a result, we derive 5 ensemble means (1 for history and 4 for future) for each variable of every model, leaving the same weight among CMIP6 models. We use the average data from 2005 to 2014 to indicate conditions at present day and that from 2090 to 2099 as the future

period. We project the changes in dust cycle using the multi-model ensemble median values between future and present day, and explore the causes of changes by linking the simulated dust cycle with meteorological variables from individual models.

2.2 Measurement data

We use dust concentrations observed at 18 ground sites operated by University of Miami to validate dust concentrations at the lowest level of the 14 models. All these sites are located on the islands with 7 in the Atlantic, 7 in the Pacific, 3 in the Southern Ocean, and 1 in the Indian Ocean. Most of these sites were built near the dust source regions with the longest period of 17 years. Although the observed data are not continuous at all sites, they provide the most valuable spatiotemporal information of global dust concentrations and have been widely used in the evaluations of dust models (Ginoux et al., 2001; Yue et al., 2009; Wu et al., 2020). We also use the monthly AOD measurements from the Aerosol Robotic Network (AERONET) to validate CMIP6 models. Observed AOD is affected by many different components in addition to dust aerosols. We select a total of 19 sites with at least one-year records and the simulated DOD-to-AOD ratio larger than 0.6 as indicated by the ensemble of CMIP6 models. In this way, AOD at the selected AERONET sites is more likely dominated by dust aerosols.

2.3 Dust emission schemes

The vertical emission flux $F_i$ for a specific dust size bin $i$ in most of climate models can be derived using the generic equation:

$$F_i = C \cdot \rho_d \cdot E \cdot f_m \cdot \alpha \cdot M_i \qquad (1)$$

Here, $C$ is a tunable parameter set to derive the reasonable dust climatology in individual models. $\rho_d$ is the density of dust particle. $E$ is the impetus composed of wind friction speed ($U_f$) above the threshold values ($U_{*t}$) for saltation. The value of $U_{*t}$ is dependent on soil moisture. $f_m$ is the erodibility potential of bare soil suitable for dust mobilization, which is usually parameterized as the cover fraction of a grid cell excluding snow, ice, lake, and vegetation. $\alpha$ is sandblasting mass efficiency related to

clay fraction (%clay). $M_i$ is the mass distribution of the specific dust size bin $i$. The detailed parameterizations for each component of Equation (1) are shown for 5 selected models in Table 2. In general, the main factors influencing dust emissions include wind friction velocity, threshold wind speed, soil moisture, clay content, soil bareness, and dust particle size. These variables are used either as individual factors or in multiple components of Eq. 1. For example, in CESM2-WACCM, CESM2, NorESM2-LM, and UKESM1-0-LL, the clay fraction is used to calculate both sandblasting mass efficiency and the threshold of wind friction speed (Lawrence et al., 2019). In CNRM-ESM2-1, $f_m$ and $\alpha$ are combined to calculate $U_{*t}$ rather than acting as individual factors in the emission function (Zakey et al., 2006).

**3 Results**

3.1 Model validations

Fig. 1a shows the spatial distribution of ground-based sites for dust observations. These sites cover a wide range of oceanic areas with different distances to source regions. Compared to observed concentrations (Fig. 1b), the simulations yield correlation coefficients (R) of 0.30-0.88 for 14 climate models, among which 12 models show R of higher than 0.8 (Table S2). Meanwhile, the simulations show normalized standard deviations (NSD, standard deviation of the model divided by that of the observations) ranging from 0.07 to 2.16. Compared to observed AOD (Fig. 1d), the simulations yield R of 0.26-0.79 and NSD of 0.28-0.95 (Table S2). With the validations, we select 9 models for the future projections including CESM2-WACCM, CESM2, CNRM-ESM2-1, GFDL-ESM4, GISS-E2-1-G, GISS-E2-1-H, GISS-E2-2-G, NorESM2-LM and UKESM1-0-LL. All of these selected model yield NSD between 0.25 and 1.5 and correlation coefficients higher than 0.55 against observations of both dust concentrations and AOD.

The ensemble mean of dust concentrations from 9 selected CMIP6 models is compared to observations at individual stations (Fig. 2). The models reproduce observed magnitude at 6 sites (Figs 2a-2f) downwind of Saharan dust sources with relative mean biases (RMB) ranging from -40% to 37.4%. For these sites, the model

ensemble also captures reasonable dust seasonality except for the underestimation of peak values in summer for Barbados (Fig. 2a) and those in spring for Cayenne (Fig. 2b). For the rest sites, the multi-model ensemble prediction overestimates dust concentrations at 1 site in the North Atlantic (Fig. 2g), 3 sites in the southern ocean (Figs 2h-2j), and 3 sites in the central Pacific (Fig. 2k-2m), most of which are far away from dust source regions. In contrast, model simulations underestimate dust concentrations at 1 site in the Indian Ocean (Fig. 2n) and 2 sites at the offshore of East Asia (Figs 2o-2p). In sum, the simulated dust concentrations show smaller spatial gradients than observations.

The ensemble mean of AOD from 9 selected CMIP6 models is compared to observations at 19 AERONET stations (Fig. 3). For six sites (1-6) in the inner North Africa, the model prediction underestimates observed peaks in springtime, especially at Bidi Bahn and Djougou. As a result, the ensemble predictions at these sites are lower than observations by at least -20% except for DMN Maine Soroa. For three sites (7-9) along the western coast of North Africa, the model ensemble captures the summertime maximum but tends to slightly overestimate AOD in other seasons. For 9 sites (10-18) in Middle East, the predicted AOD reproduces observed seasonality and magnitude with RMB between -27.7% and 20.7%. However, for the only site (CASLEO) in South America, the model prediction shows much higher AOD than measurements. The validations show that simulated AOD from the selected CMIP6 models agree well with the observed spatial pattern especially at regions near dust sources.

3.2 Dust cycle at present day

Based on the selected models, the ensemble median dust emissions, concentrations, and depositions are assessed for 2005-2014 (Fig. 4). About 87% of dust emissions are located in the Northern Hemisphere, with hotspots over North Africa, Middle East, West Asia, and Taklimakan and Gobi Deserts (Fig. 4a). The source intensity is much smaller in the Southern Hemisphere, with moderate emissions over Australia, South Africa, and southern South America. The global total dust emission from the ensemble of models is about 2566±1996 Tg, to which the emissions from Africa alone contribute

by 67 % (Table 3). Three (CESM2, CESM2-WACCM, and NorESM2-LM) out of nine models show scattered emissions while the rest show more continuous distribution (Fig. S1).

The spatial distribution of dust deposition resemble that of emissions but with much larger coverage. Dry deposition is usually confined to the source regions (Fig. 4c) because dust particles with large size are more likely to settle down and cannot travel far away from the source. In contrast, wet deposition is more dispersed (Fig. 4d) because small particles can be transported long distances to the downwind areas and finally washed out by rain. On the global scale, the annual total dry deposition is $1749\pm1919$ Tg, more than two times of the $796\pm372$ Tg by wet deposition.

The dust budget (emission minus deposition) shows net sources of $386\pm87$ Tg a$^{-1}$ in Africa and $77\pm32$ Tg a$^{-1}$ in Asia (Table 3 and Table S3). Meanwhile, the ocean acts as a net sink with the largest strength of $-250\pm62$ Tg a$^{-1}$ in the Atlantic and the secondary of $-117\pm47$ Tg a$^{-1}$ in the Indian Ocean due to their vicinity to the source regions on the land. Following the emission pattern, dust loading shows high values ($>120$ mg m$^{-2}$) around the source regions especially North Africa and decreases gradually towards global oceans (Fig. 4b).

3.3 Projection of future dust emissions

We calculate the changes of dust emissions at the end of the 21$^{st}$ century (2090-2099) relative to the present day (2005-2014). Global total emissions increase under three scenarios, with the largest change of $60.7\pm542$ Tg a$^{-1}$ (5.0%) in the SSP5-8.5 scenario (Fig. 5d). However, the total emissions show a moderate reduction of $-5.6\pm503$ Tg a$^{-1}$ (-0.46%) in the SSP3-7.0 scenario (Fig. 5c). The most significant changes are located at the major dust source regions, such as North Africa, Taklimakan and Middle East. Dust emissions in North Africa increase in all four scenarios, though with regional heterogeneous responses and varied magnitude of 4.8-47.4 Tg a$^{-1}$ (0.6%-5.6%) (Table 4). The secondary enhancement is found at Australia with increases of 1.1-4.3 Tg a$^{-1}$ (2.8%-10.7%) except SSP3-7.0 scenario (Table 4). In contrast, dust emissions in Taklimakan and Gobi Deserts show decreases of -0.4 to -6.2 Tg a$^{-1}$ (-0.8% to -11.9%),

which are stronger than the enhancement in North Africa under the SSP3-7.0 scenario
(Table 4). Furthermore, dust emissions over Asia (including Taklimakan, Gobi Deserts,
West Asia and Middle East) decrease in most scenarios especially for SSP3-7.0, in
which the regional reduction causes the global decline of dust emissions (Fig. 5c). The
inter-model variability is much higher than the projected median changes, suggesting
the large uncertainties among climate models.
We further explore the associated changes in meteorological conditions at the
source regions (Fig. 6). For North Africa, regional precipitation shows mild reductions
under all four scenarios even though the baseline rainfall is very low. The ensemble
projections show decreased relative humidity of -0.6% to -3.0% and increased surface
wind speed of 0.01-0.08m s$^{-1}$ over North Africa for all scenarios, contributing to the
largest enhancement of regional dust emissions. Similarly, projections show decreased
precipitation and relative humidity but increased surface wind over South Africa,
resulting in the increase of local emissions. As a comparison, precipitation, relative
humidity, and surface wind all show decreasing trends in Australia, where the dust
emissions increase for most scenarios except SSP3-7.0. It indicates that the effect of
drier conditions overweighs the decreased momentum for dust emissions in this specific
region. Among the total of 18 region labels (the red labels on Fig. 6) with increased
dust emissions under the four scenarios, 14 labels show decreased relative humidity by
at least 0.5%, 14 labels show decreased precipitation, and 10 labels show increased
wind speed.
In contrast, the future dust emissions decrease in Taklimakan, Gobi Deserts,
Middle East and West Asia under most scenarios (Fig. 5). Climate projections show
increased precipitation (Fig. S6) and relative humidity (Fig. S7), but decreased wind
speed (Fig. S8) over the source regions in Taklimakan and Gobi Deserts. All these
changes in meteorological conditions tend to inhibit regional dust mobilization. The
most significant reduction of 11.9% occurs in SSP3-7.0 scenario, in which regional
precipitation increases by 0.14 mm day$^{-1}$, and surface wind speed decreases by 0.08 m
s$^{-1}$. For Middle East and West Asia, the slight increase of precipitation (Fig. 6)
overweighs the moderate increase of surface wind speed, leading to a decline of
regional dust emissions for SSP1-2.6 and SSP2-4.5 (Fig. 6). Specifically, almost all the
10 region labels with reduced dust emissions under the four scenarios show increased
regional precipitation but decreased wind speed, though 8 labels show decreased
relative humidity (Fig. 6). It suggests that changes in precipitation and wind speed play
more dominant roles in the changes of dust emissions.
We select four main source regions where dust emissions are projected to increase
by at least 1 Tg a$^{-1}$ under most of future climatic scenarios (Table 4). In these regions,
we quantify the sensitivity of dust emissions to perturbations in meteorological factors
(Fig. 7). We find positive correlations between the changes in dust emissions and that
of wind speed for all models and scenarios. The largest correlation coefficient of 0.68
is derived over Taklimakan and Gobi Deserts (Fig. 7b). In contrast, precipitation is
negatively correlated with dust emissions across models and scenarios (Fig. 7). On
average, we derive the increases of dust emissions by 33.1-123.3 Tg per 0.1 m s$^{-1}$
increase in surface wind (Figs 7a-7d), and 9.6-365.0 Tg per 0.1 mm day$^{-1}$ reduction in
precipitation (Figs 7e-7h) over the main dust source regions based on the multi-model
ensemble projections. Following these sensitivities, the inter-model spread of
meteorological changes leads to the large uncertainties in the projection of future dust
emissions. Among the nine climate models, UKESM1-0-LL shows the largest
reductions of wind speed while the highest enhancement of precipitation in most of
source regions, resulting in the largest decline of dust emission for this model under all
the four scenarios (Fig. 7). In contrast, CNRM-ESM2-1 exhibits the largest increase of
wind speed and the consequent enhancement of dust emissions in North Africa.
Meanwhile, CESM2-WACCM yields the highest enhancement of dust emissions in
Australia where this model projects a protruding reduction of precipitation.

3.4 Projection of future dust loading
The dust column loading show more continuous changes than dust emissions (Fig.
8). By the end of the 21$^{st}$ century, dust loading increases along the "North Africa-
Atlantic-North America" and "Australia-South Africa-South America" belts, but
decreases along the "central Asia-East Asia-North Pacific" belt. Such pattern is in

general consistent among all four future scenarios with the strongest magnitude under the SSP5-8.5 scenario. The loading in Middle East and West Asia shows mixed responses with increasing trend in the SSP5-8.5 scenario but decreasing trends in other scenarios. In sum, dust loading increases by 0.1-668.3 Gg (1.0%-13.5%) with enhancement of column load in most regions except for Asia and its downwind regions (Fig. 8 and Table S4).

We select four dust source regions and two non-source areas in Asia to analyze the driving factors for the changes in dust loading (Fig. 9). Analyses show positive correlation coefficients ranging from 0.72 to 0.90 between dust loading and emissions. In contrast, negative correlations from -0.12 to -0.68 are yielded between the loading and precipitation. The higher magnitude of correlations in the former relationship suggests that the changes of emission dominate the variations of dust loading. However, the role of precipitation cannot be ignored as it can magnify the impact of emissions. For example, dust emissions in the source region of South Africa increase by 2.1%-10.3% under different scenarios (Table 4), while dust loading in this region increases by 2.2%-38.3% (Table S4). The higher enhancement of dust loading than emissions is mainly attributed to the decreased precipitation (Fig. S6), which reduces the proportion of wet deposition to the total deposition (Fig. S9).

For the non-source areas such as East Asia and South Asia, the moderate changes of dust emissions cannot explain the significant reductions in dust loading. Instead, the strong enhancement of regional precipitation (Fig. S6) helps promote wet deposition of dust in Asia, leading to the reduced amount of suspended particles (Fig. 8) and the increased percentage of wet-to-total deposition (Fig. S9). Studies have projected that global warming tends to enhance East Asian summer monsoon and South Asian summer monsoon, leading to increased precipitation in the middle and low latitudes of Asia (Sabade et al., 2011; Wang et al., 2018; Wu et al., 2022). These changes are not favorable for regional dust mobilization but tend to decrease dust loading through increased wet deposition.

**4 Conclusions and discussion**

Based on the multi-model ensemble approach, our study projected the changes of
dust emissions and loadings by the end of 21$^{st}$ century relative to present day. It is found
that dust emissions likely increase in Africa and Australia but decrease in Asia. Such a
pattern is consistent among different climate scenarios though the magnitude of
regional changes show some variations. As a result, the net changes of global dust
emissions vary among future scenarios with the moderate changes in SSP3-7.0 due to
the strongest emission reduction over Asia, but the large increase of 5.0% in SSP5-8.5
because of the prominent dust emission enhancement in Africa. The changes of dust
loading in general follow that of emissions but with joint impacts of precipitation,
which affects the loading through wet deposition. The decreased precipitation may
further promote dust loading over regions with increased emissions (e.g. South Africa)
through the reductions in wet deposition. In contrast, increased precipitation decreases
dust loading by more wet deposition over regions with moderate or limited changes in
dust emissions (e.g., East Asia).
Our projection revealed large uncertainties in the future global dust cycle. These
uncertainties are firstly originated from the discrepancies in the dust emission schemes
and the size bins/ranges employed by different climate models. To limit the negative
impacts of model diversity, we validated the simulated low-level dust concentrations
and AOD, and selected the models with reasonable performance. The ensemble mean
of these selected models could better capture the observed magnitude and distribution
of dust concentrations and AOD (Figs 2 and 3). However, such validations excluded
several available models, potentially increasing the uncertainties of multi-model
ensemble due to the small sample size. Based on the recent evaluations (Wu et al., 2022;
Zhao et al., 2022), the latest version of CMIP models did not improve the performance
in the simulated dust cycles, including concentrations, deposition, and optical depth,
suggesting that the more validations may rule out even more available models for the
future projection. As a result, the observation-based constraint of emission schemes (e.g,
adjusting the tunable parameter $C$ in Equation 1) and size bins (e.g., extending or
reducing the size range) in individual models is a requisite step to reduce the
uncertainties in modeling the global dust cycle.
For this study, we did not validate the long-term trend of simulated dust variables
due to the data limitations. A recent work by Kok et al. (2023) showed increasing global
dust loading during historical periods with the glacier deposition records and found that
all the CMIP6 models could not reproduce such tendency. While this newly derived
dataset provides a unique aspect for global dust activity, more validations are required
using the ground-based concentrations and/or satellite-retrieved AOD. For example, the
long-term records in China showed a decreasing trend of dust storm in East Asia during
1954-2000 (Wang et al., 2005), inconsistent with the upward trend in the same region
as revealed by Kok et al. (2023). Another limitation is that we ignore the possible
impacts of vegetation changes on the future dust activity. Previous studies have revealed
that dynamic vegetation process could significantly alter future dust activity
(Woodward et al., 2022). However, we were not able to identify such effects because
CMIP6 models do not output the information of dust sources and their strength. As a
check, we compared the changes of dust emissions at vegetation-free grid points for
both historical and future periods so as to exclude the impacts of vegetation changes.
We found very limited differences for those grids (Table S5) relative to the changes for
all grids (Table 4), suggesting that the changes of dust area are limited in most of the
CMIP6 models.
We applied the multi-model ensemble approach to minimize the projection biases
from individual models. We used the median instead of mean values from the selected
models so that our projections reflected the tendency of the majority models rather than
that of the single model with maximum changes. At present day, the ensemble
projection reasonably captures the observed dust concentrations and AOD at most sites
(Figs 2-3). The predicted annual dust emissions of 2566±1996 Tg is close to the
estimate of 2836 Tg a$^{-1}$ using an ensemble of five different dust models (Checa-Garcia
et al., 2021). The largest emission from Africa accounts for 67% of the global emissions,
similar to the estimates by previous studies (Wu et al., 2020; Aryal and Evans, 2021;
Zhao et al., 2022). The global burden of 22±8 Tg is close to the range of 12-25 Tg
estimated by Zhao et al. (2022) using three different datasets. For the future, our
ensemble projected increases of dust emissions in North Africa and Australia while the

reductions in central Asia are consistent with the results predicted using two different models (Tegen et al., 2004). The ensemble projections with the 9 selected models (Table 4) are in general consistent with the projections using all 14 models (Table S6), especially for the enhancement of dust emissions in the North Africa under all scenarios. However, both projections revealed large inter-model variability that may dampen the significance of the predicted changes.

Our sensitivity analyses showed consistent dependence of dust emissions and loadings to meteorological variables among models and scenarios (Figs 7 and 9). With such physical constraints, the trends of dust emissions are determined by the changes of regional to global meteorological fields, especially wind speed and precipitation. For example, models show contrasting tendencies of surface wind over North Africa (Fig. 7a) and precipitation in Middle East and West Asia (Fig. 7g), leading to large inter-model variability with opposite signs for the changes in dust emissions by the end of $21^{st}$ century. Given the importance of climatic change, we checked the ensemble changes in precipitation (Fig. S10) and surface wind speed (Fig. S11) with all available CMIP6 models (32 models as listed in Table S7). We found that the main features of increased drought and wind speed over North Africa and South Africa while enhanced rainfall over Asia was retained, following the "drier in dry and wetter in wet" pattern due to the land-air interactions through water and energy exchange (Feng and Zhang, 2015). It indicates that the main patterns of the changes in both dust emissions and loadings in our projections are solid. As a result, we suggest that dust emissions over the main source regions will likely enhance in a warming climate, contributing to the increased dust aerosol particles and radiative perturbations by the end of the $21^{st}$ century.

**Data availability.** The model output data from CMIP6 were downloaded from https://esgf-node.llnl.gov/search/cmip6/.

**Author contributions.** XY conceived the study. XY and YZ designed the research, conducted the data analysis and paper writing. YaC, JZ, CT and HZ provided paper writing advices and helped with data analysis procedures. YuC, YH, WF and XZ

provided scientific advices. All co-authors contributed to improve the manuscript.

**Competing interests.** The contact author has declared that none of the authors has any
competing interests.

**Acknowledgements.** This research was supported by the National Key Research and
Development Program of China (grant no. 2019YFA0606802).

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

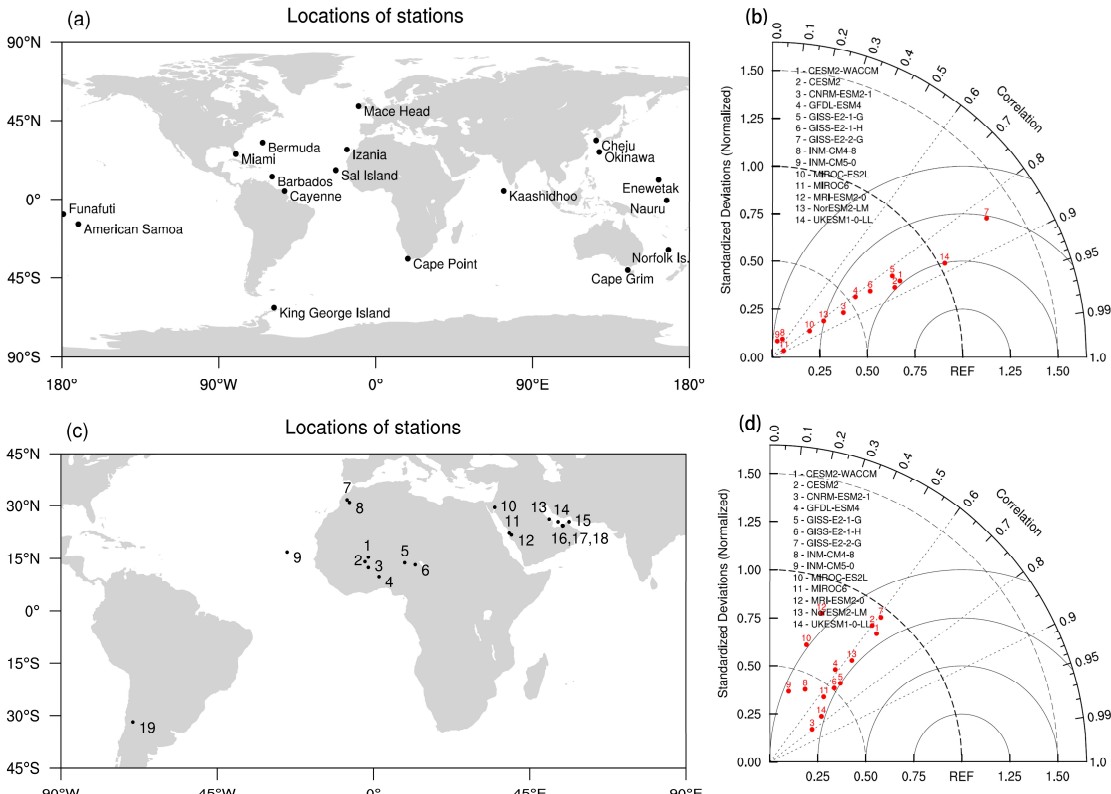

**Figure 1.** (a) Locations of 18 observational stations in the University of Miami Ocean Aerosol Network and the (b) evaluation of simulated dust concentrations from CMIP6 models at these stations. (c) Locations of 19 AERONET sites and the (d) evaluation of simulated AOD from CMIP6 models at these stations. The names of AERONET sites in (c) are 1-Agoufou, 2-Bidi_Bahn, 3-Ouagadougou, 4-Djougou, 5-Zinder_Airport, 6-DMN_Maine_Soroa, 7-Ras_El_Ain, 8-Ouarzazate, 9-Calhau, 10-Eilat, 11-KAUST_Campus, 12-Hada_El-Sham, 13-Bahrain, 14-Abu_Al_Bukhoosh, 15-Dhadnah, 16-Mussafa, 17-Dhabi, 18-Masdar_Institute, 19-CASLEO. The longitudes and latitudes of these sites are indicated on Figures 2 and 3.

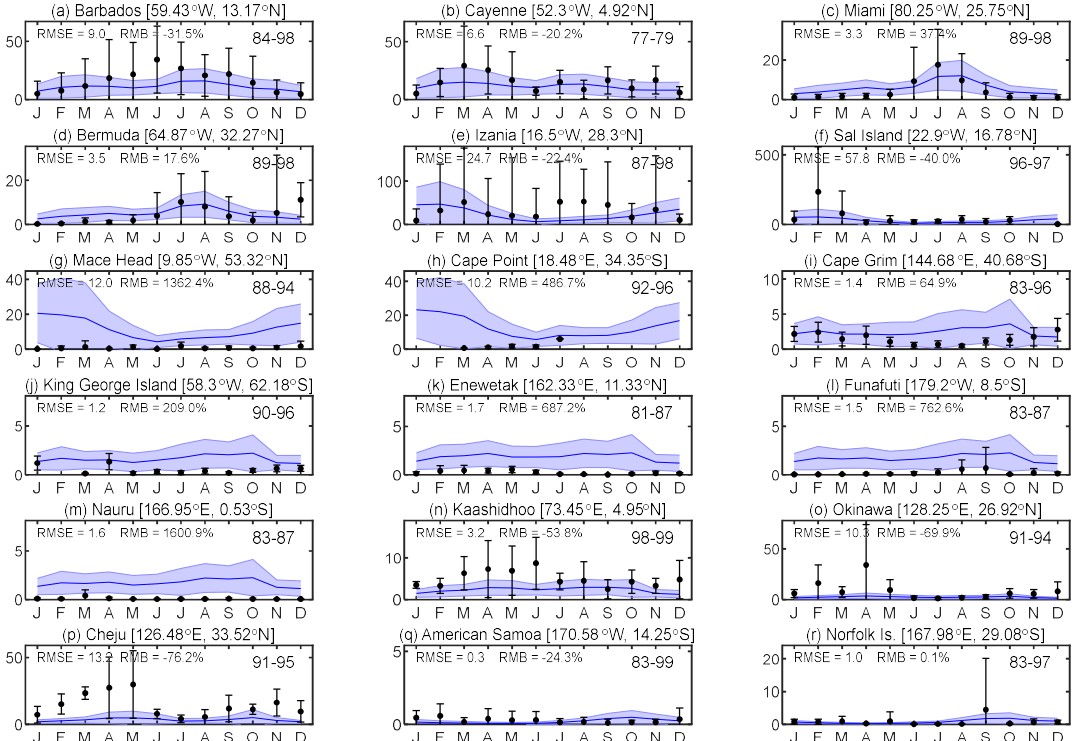

**Figure 2.** Comparison of monthly dust concentrations (units: µg m⁻³) between ensemble

simulations by CMIP6 models and observations at 18 sites. The solid lines represent

ensemble mean of simulations with shadows indicating inter-model spread. The points

are the monthly mean of observations with errorbars indicating year-to-year variability.

The time span of observations at each site is shown in the upper right corner of each

panel. Root mean square error (RMSE) and relative mean biases (RMB) of observations

and simulations are shown in the upper left corner of each panel.



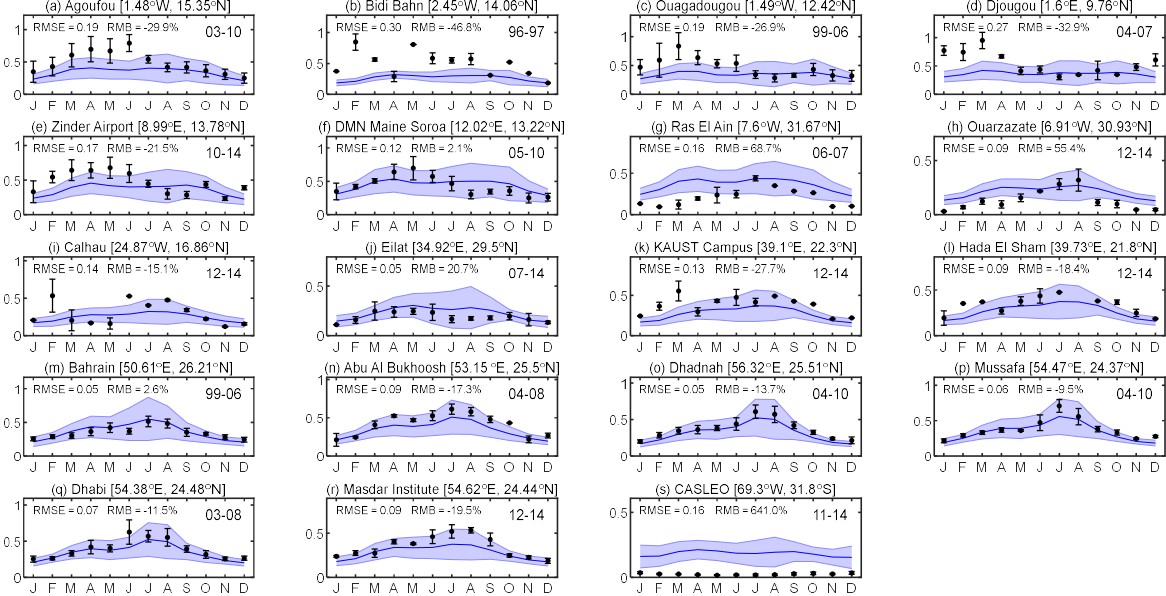


Figure 3. The same as Figure 2 but for the validation of the ensemble simulated aerosol

optical depth at 19 AERONET sites.



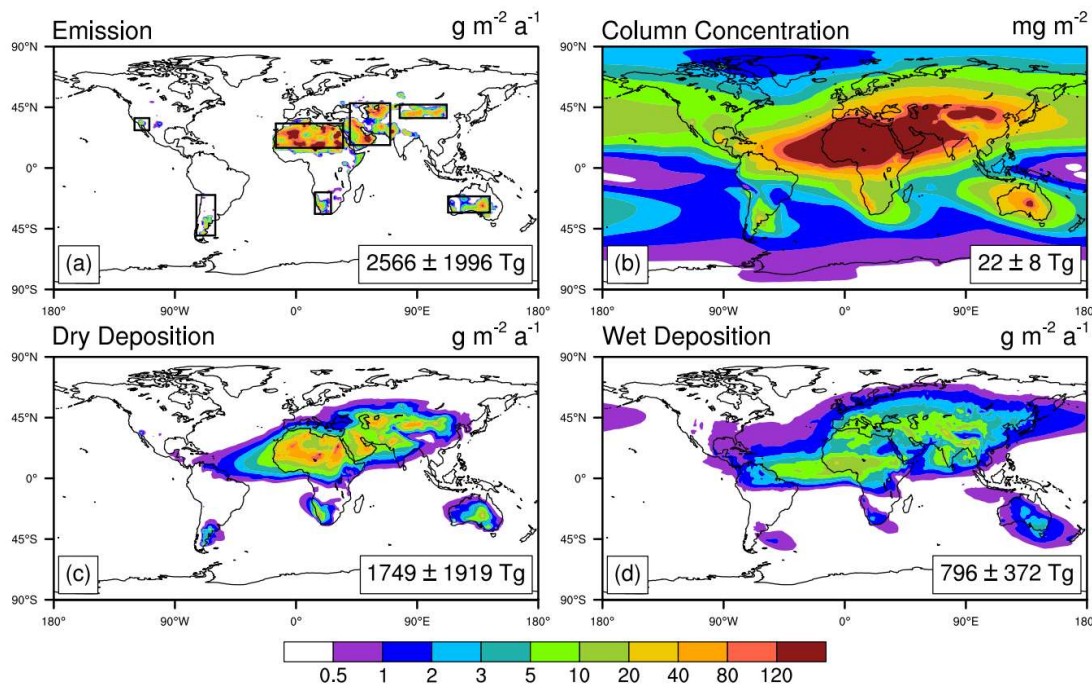

**Figure 4.** Multi-model ensemble of (a) emissions, (b) column load, (c) dry deposition, and (d) wet deposition of dust aerosols at present day (2005-2014). The box regions on (a) are dust sources of North Africa (NAF) (15°N-33°N, 15°W-35°E), Middle East and West Asia (MEWA) (17°N-48°N, 40°E-70°E), Taklimakan and Gobi Deserts (TGD) (37°N-47°N, 77°E-112°E), Australia (AUS) (33°S-21°S, 113°E-144°E), North America (NAM) (28°N-37°N, 120°W-109°W), South America (SAM) (50°N-20°N, 74°S-60°S), and South Africa (SAF) (34°S-18°S, 14°E-26°E). The detailed results for individual models are shown in Fig. S1.

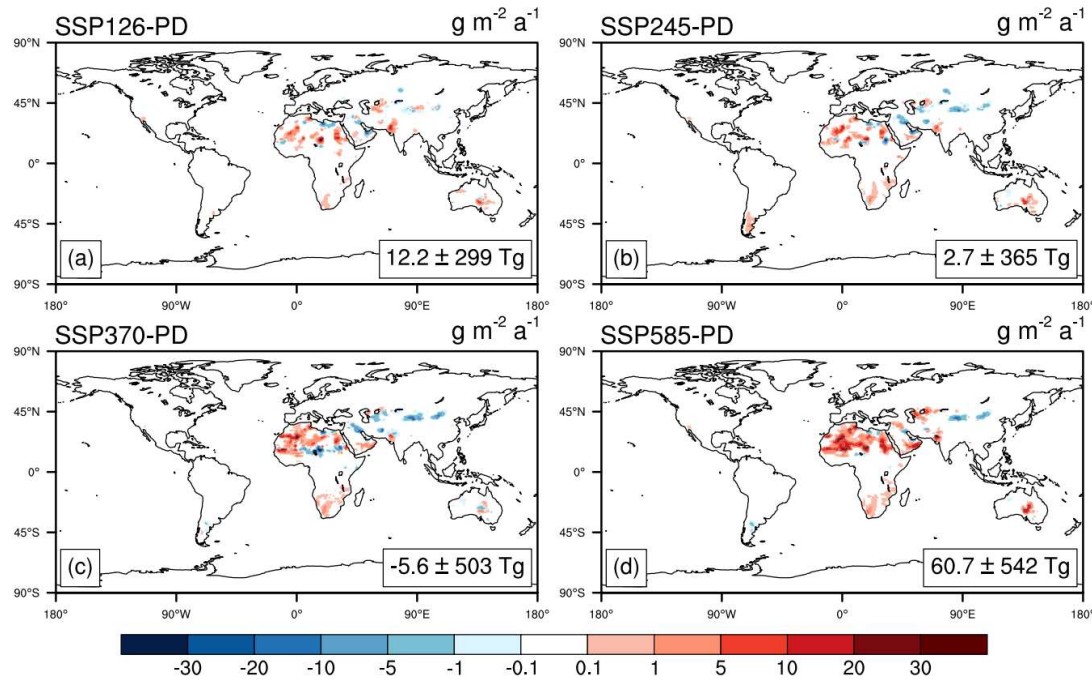

**Figure 5.** Multi-model ensemble projection of the changes in dust emissions by the end of 21st century (2090-2099) relative to present day (2005-2014) under four different anthropogenic emission scenarios. The detailed projections at 2090-2099 for individual models are shown in Fig. S2-S5 under four different scenarios.

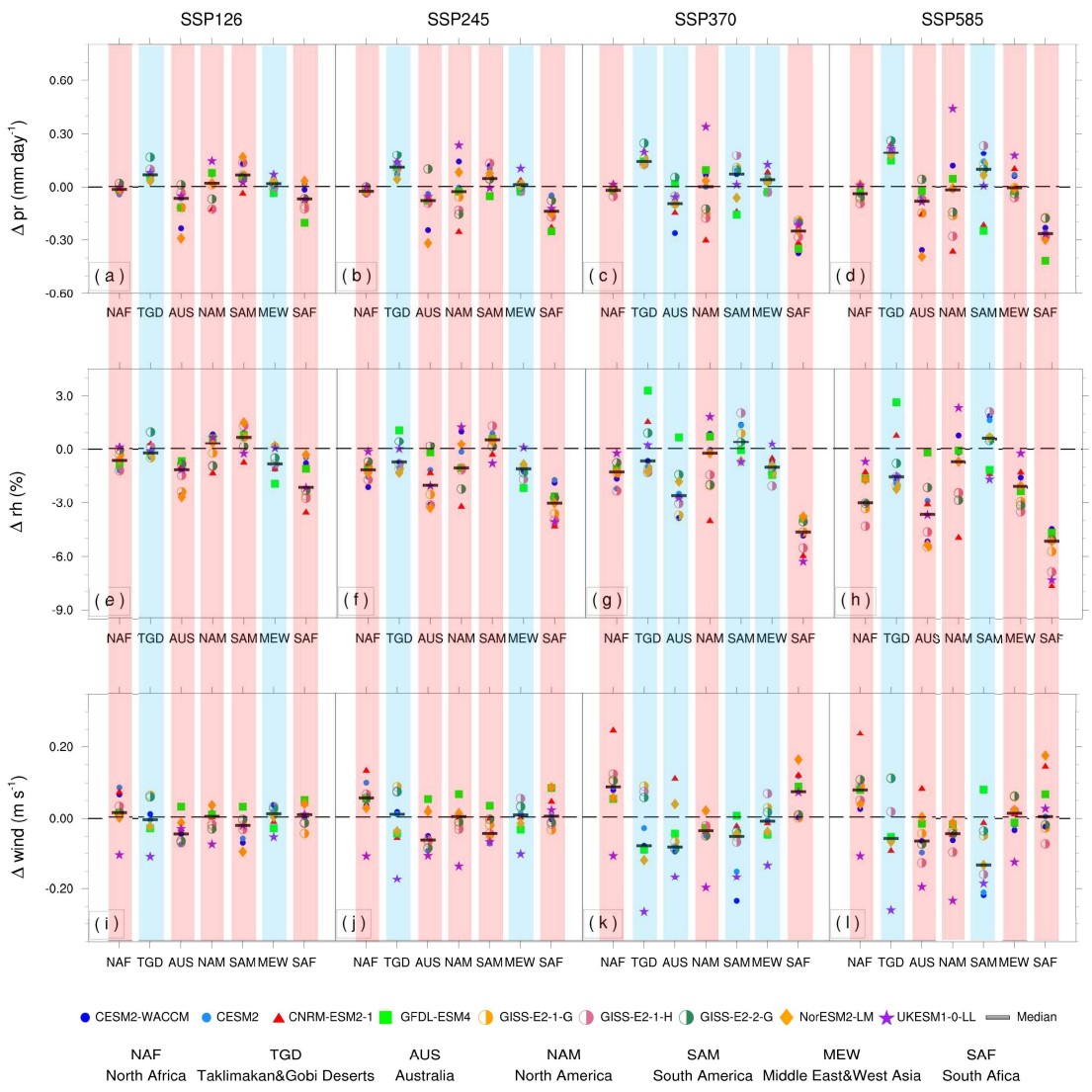

**Figure 6.** Changes of meteorological factors over main dust emission regions under four SSP scenarios by the end of 21[st] century (2090-2099) relative to present day (2005-2014). Each box column represents a future climate scenario, including SSP1-2.6, SSP2-4.5, SSP3-7.0 and SSP5-8.5. Each row represents a meteorological factor, including precipitation (top), relative humidity (middle), and surface wind (bottom). Regions with emissions increasing are marked with light red bars, while regions with emissions decreasing are marked with light blue bars.

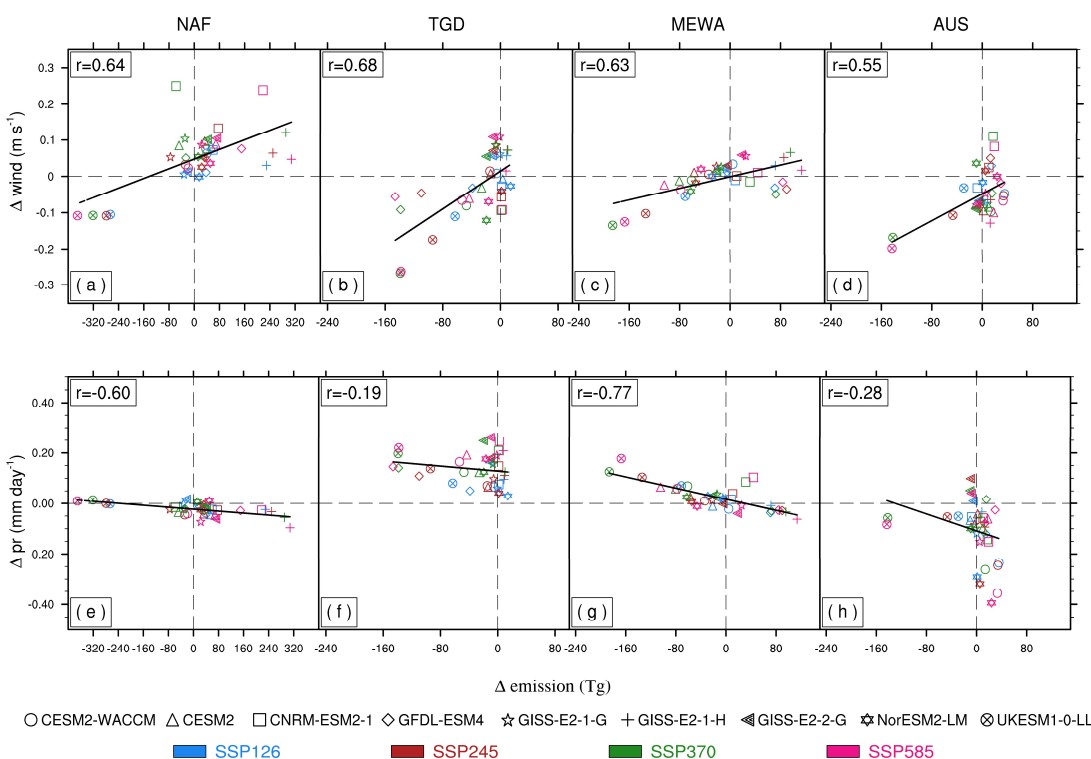

**Figure 7.** Relationships between the changes of dust emissions and the changes of meteorological factors. Each column represents a source region, including North Africa (NAF), Taklimakan and Gobi Deserts (TGD), Middle East and West Asia (MEWA), and Australia (AUS). Each row represents a meteorological factor, including surface wind (top) and precipitation (bottom).

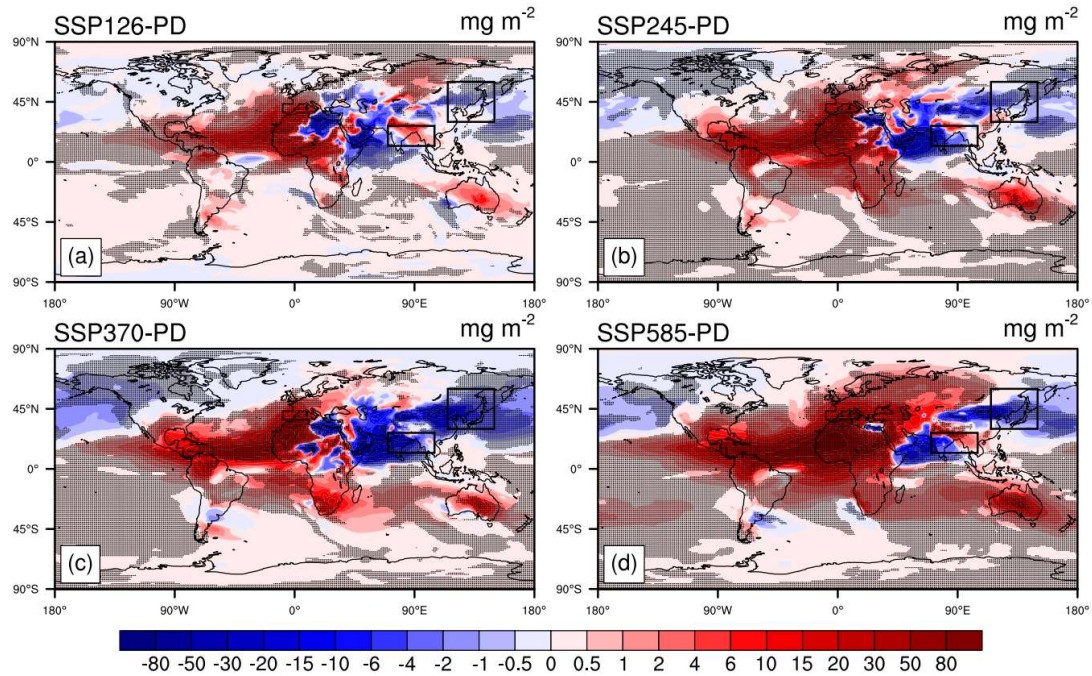

**Figure 8.** Multi-model ensemble projection of the changes in dust column load by the end of 21st century (2090-2099) relative to present day (2005-2014). Dotted areas represent changes significant at 90% level. Two additional box areas are selected for South Asia (12°N-27°N, 70°E-105°E) and East Asia (30°N -60°N, 115°E -150°E).

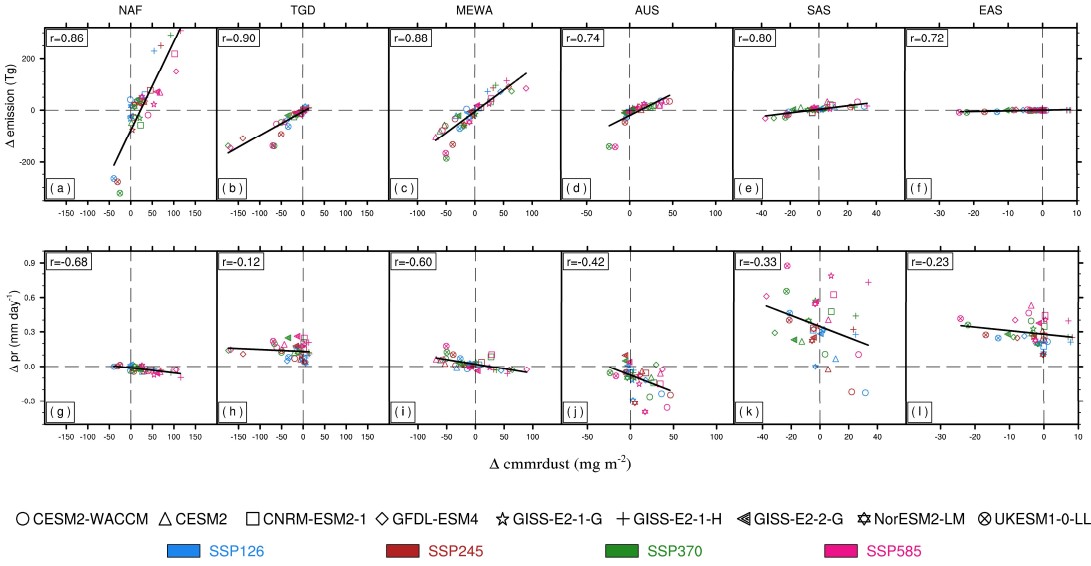

**Figure 9.** Relationships between the changes of dust column load and the change of influencing factors. From left to right, each column represents a specific region including North Africa (NAF), Taklimakan and Gobi Deserts (TGD), Middle East and West Asia (MEWA), Australia (AUS), South Asia (SAS) and East Asia (EAS). Each row represents an influencing factor, including dust emissions (top) and precipitation (bottom).

**Table 1.** The information of CMIP6 models

| Model [a] | Nation | Resolution | Number of runs for dust cycle | | | | |
|---|---|---|---|---|---|---|---|
| | | | Hist | SSP126 | SSP245 | SSP370 | SSP585 |
| **CESM2-WACCM** | U.S. | 1.25°×0.94° | 3 | 1 | 5 | 3 | 5 |
| **CESM2** | U.S. | 1.25°×0.94° | 11 | 3 | 3 | 3 | 3 |
| **CNRM-ESM2-1** | France | 1.4°×1.4° | 3 | 5 | 10 | 5 | 5 |
| **GFDL-ESM4** | U.S. | 1.25°×1° | 1 | 1 | 1 | 1 | 1 |
| **GISS-E2-1-G** | U.S. | 2.5°×2° | 19 | 10 | 25 | 17 | 10 |
| **GISS-E2-1-H** | U.S. | 2.5°×2° | 10 | 5 | 5 | 1 | 5 |
| **GISS-E2-2-G** | U.S. | 2.5°×2° | 5 | 5 | 5 | 5 | 5 |
| INM-CM4-8 | Russia | 2°×1.5° | 1 | 1 | 1 | 1 | 1 |
| INM-CM5-0 | Russia | 2°×1.5° | 10 | 1 | 1 | 5 | 1 |
| MIROC-ES2L | Japan | 2.8°×2.8° | 31 | 10 | 30 | 10 | 10 |
| MIROC6 | Japan | 1.4°×1.4° | 10 | 3 | 3 | 3 | 3 |
| MRI-ESM2-0 | Japan | 1°×1° | 12 | 5 | 10 | 5 | 6 |
| **NorESM2-LM** | Norway | 2°×2° | 1 | 1 | 13 | 1 | 1 |
| **UKESM1-0-LL** | U.K. | 1.875°×1.25° | 3 | 5 | 5 | 3 | 4 |
| **Total runs** | | | 120 | 56 | 117 | 63 | 60 |


[a] The models selected for future projections are bolded.


731                **Table 2.** The parameterization schemes of dust emission function

| Model | $E$ | $M_i$ | $f_m$ | %clay | Reference |
|---|---|---|---|---|---|
| CESM2-WACCM | $U_f{}^3(1-\dfrac{U_{*t}}{U_f})(1+\dfrac{U_{*t}}{U_f})^2$ | 3 source modes, 4 dust bins | Fraction of grid cell excluding snow, lake and vegetation; depends on liquid water and ice contents in top soil layer | Used to calculate the sandblasting mass efficiency and $U_{*t}$ | Oleson et al. (2010) Wu et al. (2016) |
| CESM2 | | | | | |
| NorESM2-LM | | | | | |
| UKESM1-0-LL | $U_f{}^3(1+\dfrac{U_{*t}}{U_f})(1-(\dfrac{U_{*t}}{U_f})^2)$ | 9 dust bins | Considering grid cell fractions of vegetation | | Woodward, (2011) |
| CNRM-ESM2-1 | | 3 dust bins | Using roughness length | Used to calculate the $U_{*t}$ | Marticorena et al. (1997) Zakey et al. (2006) Nabat et al. (2015) |
| GFDL-ESM4 | $U_f{}^2(U_f-U_{*t})$ | 5 dust bins | Using leaf area index and stem area index | / | Evans et al. (2016) Dunne et al. (2020) |
| GISS-E2 | $U_f{}^2(U_f-U_{*t})$ | 6 dust bins | / | Used to calculate the $U_{*t}$ | Ginoux et al. (2004) Bauer and Koch (2005) Kelley et al. (2020) |






**Table 3.** The summary of dust cycle at present day[*]

| Region | Emission | Dry Deposition | Wet Deposition | Budget[**] |
|---|---|---|---|---|
| | Tg a$^{-1}$ | Tg a$^{-1}$ | Tg a$^{-1}$ | Tg a$^{-1}$ |
| Africa | 1713±1288 | 1091±1235 | 236±155 | 386±87 |
| Asia | 736±458 | 432±419 | 226±161 | 77±32 |
| Australia | 165±237 | 110±211 | 20±25 | 35±13 |
| South America | 52±106 | 30±63 | 21±23 | 1±30 |
| North America | 15±27 | 13±31 | 9±20 | -6±25 |
| Europe | 5±3 | 12±4 | 34±15 | -41±19 |
| Pacific Ocean | / | 14±12 | 48±23 | -62±33 |
| Indian Ocean | / | 46±23 | 71±36 | -117±47 |
| Atlantic Ocean | / | 95±39 | 155±57 | -250±62 |
| Arctic Ocean | / | 0±0.3 | 2±1 | -3±1 |

[*] Values from individual climate models are shown in Table S3
[**] Budget = Emission - Dry Deposition - Wet Deposition





**Table 4.** Multi-model ensemble projection of the absolute (Tg a$^{-1}$) and relative
changes (%) in dust emissions by the end of this century (2090-2099)

| Region | SSP1-2.6 | | SSP2-4.5 | | SSP3-7.0 | | SSP5-8.5 | |
|---|---|---|---|---|---|---|---|---|
| | Absolute | Relative | Absolute | Relative | Absolute | Relative | Absolute | Relative |
| NAF | 10.1±121.7 | 1.2 | 5.3±131.4 | 0.6 | 4.8±148.0 | 0.6 | 47.4±178.8 | 5.6 |
| TGD | -0.4±23.5 | -0.8 | -2.5±41.3 | -4.9 | -6.2±53.6 | -11.9 | -4.6±55.7 | -8.9 |
| MEWA | -0.7±43.1 | -0.3 | -4.5±66.4 | -1.8 | -4.4±81.1 | -1.8 | 6.8±87.2 | 2.7 |
| AUS | 1.1±17.0 | 2.8 | 2.1±20.7 | 5.1 | -0.1±47.2 | -0.4 | 4.3±51.6 | 10.7 |
| NAM | 0.03±4.7 | 2.2 | 0.02±6.1 | 1.3 | 0.01±5.4 | 0.8 | 0.02±5.7 | 1.4 |
| SAM | 0.02±32.3 | 0.3 | 0.4±42.1 | 6.7 | -0.1±31.3 | -2.0 | -0.4±27.7 | -6.1 |
| SAF | 0.2±4.1 | 2.1 | 0.5±4.2 | 5.5 | 0.9±11.4 | 9.9 | 0.9±5.0 | 10.3 |

* The domain of each region is shown in Figure 1a

