# Peer review of "Multi-model ensemble projection of global dust cycle by the end of 21st century using CMIP6 data"

_Atmospheric Chemistry and Physics, 2022_

## Author Comment (AC1)

**Response to the review comment1 on acp-2022-760**

*RC1: 'Comment on acp-2022-760', Anonymous Referee #1, 03 Mar 2023*

*The manuscript by Zhao et al. aims at estimating future changes in global distributions and the budget of mineral dust aerosol by making use of results from the CMIP6 model experiments. Although this is an interesting topic and the use of CMIP results is a good starting point, the study has significant weaknesses and I cannot recommend publication in ACP.*

Response: Thank you very much for your constructive suggestions. We have made our best to revise the manuscript following your comments. In this revision, we included all the available model output from CMIP6, validated the model performance with site-level AOD, added the uncertainty ranges for the projections, and compared the future changes of dust emissions over vegetation-free grids. We hope our revisions and responses could answer your concerns.

*Major points:*
*The authors point out that taking into account the effects of changes in vegetation cover on dust emissions may play an important role but using vegetation model scenario results make the dust aerosol trends more uncertain. Indeed such changes would have a significant impact on dust emissions. The authors do not clarify to which extent vegetation cover changes were considered in the CMIP model experiments that were used for this study, especially for the future scenarios. If they want to avoid uncertainties related to potential vegetation changes by considering only the effects of wind and precipitation changes, then only in regions that are vegetation-free in both the historical and future scenarios should be considered. (see e.g Mahowald et al., 2003, Woodward et al., 2005).*

*Mahowald, N. M., and Luo, C. (2003), A less dusty future? Geophys. Res. Lett., 30, 1903, doi:10.1029/2003GL017880, 17.*
*Woodward, S., Roberts, D. L., and Betts, R. A. (2005), A simulation of the effect of climate change–induced desertification on mineral dust aerosol, Geophys. Res. Lett., 32, L18810, doi:10.1029/2005GL023482.*

Response: Thank you for your suggestions. Inclusion of dynamic vegetation is expected to be an advance for the CMIP6 models. At the starting point, we hope to include vegetation cover changes as one of the drivers for future dust cycle. However, the changes of dust source areas are not provided by CMIP6 archives. In addition, the dynamic vegetation processes may instead introduce more uncertainties for the dust projections. As a result, we made great efforts to validate the simulated dust variables (concentrations and AOD, Figures 1-3) and selected the best models for future projections. In the introduction section, we have clarified as follows: "Compared to

CMIP5 models, more dust emission schemes are coupled with dynamic vegetation in the CMIP phase 6 (CMIP6) models to optimize land surface emission processes (Zhao et al., 2022). However, such improvement may instead amplify the uncertainties of dust simulations, because the predicted vegetation change may be inconsistent with the observed tendencies (Wu et al., 2020). As a result, it is important to validate the simulated present-day dust cycle before the application of different models in the future projection (Aryal and Evans, 2021)." (Lines 100-106)

We added Table S5 to compare the changes in dust emissions on the same vegetation-free grids as suggested: "Previous studies have revealed that dynamic vegetation process could significantly alter future dust activity (Woodward et al., 2022). However, we were not able to identify such effects because CMIP6 models do not output the information of dust sources and their strength. As a check, we compared the changes of dust emissions at vegetation-free grid points for both historical and future periods so as to exclude the impacts of vegetation changes. We found very limited differences for those grids (Table S5) relative to the changes for all grids (Table 4), suggesting that the changes of dust area are limited in most of the CMIP6 models." (Lines 387-395)

**Table S5.** Multi-model ensemble projection of the absolute (Tg a$^{-1}$) and relative changes (%) in dust emissions by the end of this century (2090-2099) at vegetation-free grid points

| Region | SSP1-2.6 | | SSP2-4.5 | | SSP3-7.0 | | SSP5-8.5 | |
|--------|----------|----------|----------|----------|----------|----------|----------|----------|
| | Absolute | Relative | Absolute | Relative | Absolute | Relative | Absolute | Relative |
| NAF | 10.1$\pm$121.7 | 1.2 | 5.8$\pm$131.5 | 0.7 | 4.2$\pm$174.1 | 0.5 | 47.4$\pm$178.9 | 5.6 |
| TGD | -0.4$\pm$23.5 | -0.8 | -2.5$\pm$41.3 | -4.9 | -6.2$\pm$53.6 | -11.9 | -4.6$\pm$55.7 | -8.9 |
| MEWA | -0.7$\pm$43.1 | -0.3 | -4.5$\pm$66.4 | -1.8 | -4.4$\pm$81.0 | -1.8 | 6.8$\pm$87.2 | 2.7 |
| AUS | 1.2$\pm$16.9 | 2.9 | 2.0$\pm$20.5 | 5.1 | -0.1$\pm$47.2 | -0.3 | 4.3$\pm$51.6 | 10.7 |
| NAM | 0.03$\pm$4.7 | 2.2 | 0.02$\pm$6.1 | 1.3 | 0.01$\pm$5.4 | 0.8 | 0.02$\pm$5.7 | 1.4 |
| SAM | 0.01$\pm$32.3 | 0.2 | 0.4$\pm$42.1 | 6.7 | -0.1$\pm$31.2 | -2.1 | -0.4$\pm$27.7 | -6.2 |
| SAF | 0.2$\pm$4.1 | 2.3 | 0.5$\pm$4.1 | 6.1 | 0.7$\pm$10.6 | 9.0 | 0.9$\pm$5.0 | 11.4 |

\* The domain of each region is shown in Fig. 4a

*The selection criteria that the authors use to decide what are the 'good' models to be used in this study is vague. Only surface concentration data of mineral dust are compared with station data at several locations, which are likely covering different time periods (not given here). For several sites none of the models reproduces the observed concentrations at certain seasons, which hints toward a fundamental problem of dust modelling at global scales. Given that mineral dust concentrations are highly variable in time and space including altitude of dust transport, to show that that the dust cycle is reproduced well by the models other data such as optical thickness from satellites are*

*the AERONET network should be taken into account. It would also be interesting if the results on future trends would be different if all models would be considered.*

Response: Following this suggestion, we have added the validations of CMIP6 models with the aerosol optical depth (AOD) data from AERONET network in the revised Figure 1 and new Figure 3. We selected 'good' models with reasonable correlation coefficients and low biases against observations. We added the related descriptions as follows: "We also use the monthly AOD measurements from the Aerosol Robotic Network (AERONET) to validate CMIP6 models. Observed AOD is affected by many different components in addition to dust aerosols. We select a total of 19 sites with at least one-year records and the simulated DOD-to-AOD ratio larger than 0.6 as indicated by the ensemble of CMIP6 models. In this way, AOD at the selected AERONET sites is more likely dominated by dust aerosols." (Lines 150-156) "With the validations, we select 9 models for the future projections including CESM2-WACCM, CESM2, CNRM-ESM2-1, GFDL-ESM4, GISS-E2-1-G, GISS-E2-1-H, GISS-E2-2-G, NorESM2-LM and UKESM1-0-LL. All of these selected model yield NSD between 0.25 and 1.5 and correlation coefficients higher than 0.55 against observations of both dust concentrations and AOD." (Lines 188-193)

[Figure]

**Figure 1.** (a) Locations of 18 observational stations in the University of Miami Ocean Aerosol Network and the (b) evaluation of simulated dust concentrations from CMIP6 models at these stations. (c) Locations of 19 AERONET sites and the (d) evaluation of simulated AOD from CMIP6 models at these stations. The names of AERONET sites

in (c) are 1-Agoufou, 2-Bidi_Bahn, 3-Ouagadougou, 4-Djougou, 5-Zinder_Airport, 6-DMN_Maine_Soroa, 7-Ras_El_Ain, 8-Ouarzazate, 9-Calhau, 10-Eilat, 11-KAUST_Campus, 12-Hada_El-Sham, 13-Bahrain, 14-Abu_Al_Bukhoosh, 15-Dhadnah, 16-Mussafa, 17-Dhabi, 18-Masdar_Institute, 19-CASLEO. The longitudes and latitudes of these sites are indicated on Figures 2 and 3.

"The ensemble mean of AOD from 9 selected CMIP6 models is compared to observations at 19 AERONET stations (Fig. 3). For six sites (1-6) in the inner North Africa, the model prediction underestimates observed peaks in springtime, especially at Bidi Bahn and Djougou. As a result, the ensemble predictions at these sites are lower than observations by at least -20% except for DMN Maine Soroa. For three sites (7-9) along the coast of western Africa, the model ensemble captures the summertime maximum but tends to slightly overestimate AOD in other seasons. For 9 sites (10-18) in Middle East, the predicted AOD reproduces observed seasonality and magnitude with RMB between -27.7% and 20.7%. However, for the only site (CASLEO) in South America, the model prediction shows much higher AOD than measurements. The validations show that simulated AOD from the selected CMIP6 models agree well with the observed spatial pattern especially at regions near dust sources." (Lines 207-218)

[Figure]

**Figure 3.** The same as Figure 2 but for the validation of the ensemble simulated aerosol optical depth at 19 AERONET sites.

We added a new Table S6 to compare the projection results using the selected and all models: "The ensemble projections with the 9 selected models (Table 4) are in general

consistent with the projections using all 14 models (Table S6), especially for the enhancement of dust emissions in the North Africa under all scenarios. However, both projections revealed large inter-model variability that may dampen the significance of the predicted changes." (Lines 409-413)

**Table S6.** Multi-model ensemble projection of the absolute (Tg a$^{-1}$) and relative changes (%) in dust emissions by the end of this century (2090-2099) using all 14 CMIP6 models

| Region | SSP1-2.6 Absolute median±SD | SSP1-2.6 Relative | SSP2-4.5 Absolute median±SD | SSP2-4.5 Relative | SSP3-7.0 Absolute median±SD | SSP3-7.0 Relative | SSP5-8.5 Absolute median±SD | SSP5-8.5 Relative |
|---|---|---|---|---|---|---|---|---|
| NAF | 10.3±131.5 | 1.4 | 15.8±159.9 | 2.2 | 19.5±181.6 | 2.7 | 48.0±215.2 | 6.6 |
| TGD | 0.6±32.3 | 1.1 | -1.4±37.9 | -2.9 | -4.8±50.6 | -9.6 | -3.5±54.0 | -6.9 |
| MEWA | 1.8±68.4 | 0.9 | 0.6±69.7 | 0.3 | -1.7±68.1 | -0.8 | 5.6±93.8 | 2.8 |
| AUS | -0.6±15.5 | -1.3 | 0.2±26.5 | 0.4 | -0.3±49.4 | -0.7 | 1.9±52.7 | 4.3 |
| NAM | 0.03±3.9 | 1.8 | 0.02±5.2 | 1.3 | 0.02±5.4 | 1.6 | 0.04±6.3 | 2.6 |
| SAM | 0.4±30.6 | 5.0 | 0.6±47.6 | 7.2 | 0.2±58.6 | 1.9 | 0.1±84.5 | 0.9 |
| SAF | 0.02±4.6 | 0.2 | 0.7±5.2 | 7.8 | 1.3±10.2 | 13.8 | 1.5±7.8 | 15.9 |

* The domain of each region is shown in Fig. 4a

*Another point regarding the selection of models used in this study: Rather than comparing multi-year average concentrations, the decadal temporal trends in dust aerosols simulated by the models would give a better indication for their suitability of predicting future changes (eg., Kok et al 2023).*

*Kok, J.F., Storelvmo, T., Karydis, V.A. et al. Mineral dust aerosol impacts on global climate and climate change. Nat Rev Earth Environ **4**, 71–86 (2023). https://doi.org/10.1038/s43017-022-00379-5*

Response: Thank you for the suggestion. We agree that it's important to validate the long-term trend of dust aerosols in CMIP6 models. However, such validation is difficult due to the limitations in credible datasets. The recent loading dataset developed by Kok et al. (2023) is very unique but also needs further comparisons with other data sources. For example, the increasing trend revealed by Kok et al. (2023) was not consistent with the decreasing trend of dust storms in Asia (Figure R1) (Wang et al., 2005). We expect that the long-term trends of simulated dust aerosols could be further evaluated with more available observations in the future studies.

[Figure]

Fig. 4. Interannual variation of annual occurrence days of three kinds of dust storm events in China during 1954–2000.

Figure R1. Dust storm record from (Wang et al., 2005)

In the revised paper, we clarified as follows: "For this study, we did not validate the long-term trend of simulated dust variables due to the data limitations. A recent work by Kok et al. (2023) showed increasing global dust loading during historical periods with the glacier deposition records and found that all the CMIP6 models could not reproduce such tendency. While this newly derived dataset provides a unique aspect for global dust activity, more validations are required using the ground-based concentrations and/or satellite-retrieved AOD. For example, the long-term records in China showed a decreasing trend of dust storm in East Asia during 1954-2000 (Wang et al., 2005), inconsistent with the upward trend in the same region as revealed by Kok et al. (2023)." (Lines 378-386)

*As shown in Figures S7-S11 in the supplemental material, the results for emission and deposition in selected individual models are differ greatly from each other (dust column loads should be added as additional figure). Ranges need to be given for all results. In those places where ranges are provided for scenario results (e.g. page 8 line 224 to 227) it is unclear what the range refers to – Standard deviations? Results from different Models?*

Response: We added the standard deviations among different models as the inter-model variability for all results in Figures (Figs 4-5) and Tables (Table 3-4 and Table S4-S6). We also discussed the possible impacts of inter-model variability on the projections: "The inter-model variability is much higher than the projected median changes, suggesting the large uncertainties among climate models." (Lines 261-263) "However, both projections revealed large inter-model variability that may dampen the significance of the predicted changes." (Lines 412-413)

*What is the reasoning behind focusing on relative humidity in addition to wind speed and precipitation as the main factors influencing dust trends? Relative humidity does not impact the dust cycle directly. Also in Figure 6 there appears to be a better*

*correlation for precipitation than relative humidity anyway. If the relative humidity is supposed to represent drought condition one could instead use.*

Response: We agree that relative humidity is less important than precipitation in modulating dust emissions. Our analyses of Figure 6 further confirm this point: "Specifically, almost all the 10 region labels with reduced dust emissions under the four scenarios show increased regional precipitation but decreased wind speed, though 8 labels show decreased relative humidity (Fig. 6). It suggests that changes in precipitation and wind speed play more dominant roles in the changes of dust emissions." (Lines 288-292)

In the revised paper, we have removed the relationships between relative humidity and dust emissions in Figure 7:

[Figure]

**Figure 7.** Relationships between the changes of dust emissions and the changes of meteorological factors. Each column represents a source region, including North Africa (NAF), Taklimakan and Gobi Deserts (TGD), Middle East and West Asia (MEWA), and Australia (AUS). Each row represents a meteorological factor, including surface wind (top) and precipitation (bottom).

*In Figure 7 it hard to see the dotted areas which indicate significant changes in the figure. (I recommend to mask out areas with non-significant changes eg. by grey color). In any case, it appears that the focus regions downwind of East Asia do not contain significant changes and thus should not be highlighted in the paper.*

Response: We bolded the points in the Figure 8 (original Figure 7). We mainly discussed the results with significant increases (South Africa) and reductions (East Asia) of dust

loading: "the role of precipitation cannot be ignored as it can magnify the impact of emissions. For example, dust emissions in the source region of South Africa increase by 2.1%-10.3% under different scenarios (Table 4), while dust loading in this region increases by 2.2%-38.3% (Table S4). The higher enhancement of dust loading than emissions is mainly attributed to the decreased precipitation (Fig. S6), which reduces the proportion of wet deposition to the total deposition (Fig. S9)." (Lines 330-335) "For the non-source areas such as East Asia and South Asia, the moderate changes of dust emissions cannot explain the significant reductions in dust loading. Instead, the strong enhancement of regional precipitation (Fig. S6) helps promote wet deposition of dust in Asia, leading to the reduced amount of suspended particles (Fig. 8) and the increased percentage of wet-to-total deposition (Fig. S9)." (Lines 336-340)

[Figure]

**Figure 8.** Multi-model ensemble projection of the changes in dust column concentrations by the end of 21$^{st}$ century (2090-2099) relative to present day (2005-2014). Dotted areas represent changes significant at 90% level. Two additional box areas are selected for South Asia (12°N-27°N, 70°E-105°E) and East Asia (30°N -60°N, 115°E -150°E).

*High resolution convection-resolving results show that wet convection driven dust emissions (cold pools) cannot be represented correctly, and that strengthening convective activity in future scenarios may enhance dust emissions in the southern Sahara in NH summer, but may lead to overestimating of low-level jet emissions (Garcia Carreras, 2021). It is questionable to which extent the results of coarse-resolved global models such as used in CMIP are suitable future change estimates, at least in regions that are strongly affected by convective activity.*

*Garcia-Carreras, L., Marsham, J.H., Stratton, R.A. et al. Capturing convection essential for projections of climate change in African dust emission. npj Clim Atmos Sci 4, 44 (2021). https://doi.org/10.1038/s41612-021-00201-x*

Response: It's a good question about the model capability in future projections. The coarse model resolution indeed hinders the prediction of convective activity, which may alter the dust emissions. However, there are no enough data to validate the model performance in capturing the convective activity and the consequent effects on dust emissions on the global scale. For this study, we have to use the up-to-date most advanced dust models coupled in climate models. We also applied the ensemble approach to minimize the impact of inter-model variability. Our site-level validations using concentrations and AOD demonstrate the capability of CMIP6 models. We believe our study provides the most reliable projections of future dust cycle based on our current understandings of dust schemes and the available observations.

*Minor points:*

*The CMIP experiments should be explained in some more detail. Why is only one ensemble member selected for each model? Why is the analysis limited to 10 years?*

Response: In the revised paper, we added all available models and ensemble runs for future projections. This largely increases our sample number from the original 50 (10 models with one run for each of 5 climatic scenarios) to 416 (14 models with varied runs under 5 climatic scenarios) for each dust variable. We limited our analysis to 10 years for each scenario just to estimate the decadal means. The choice of time period length will not change our main conclusions.

In the revised paper, we clarified as follows: "We use all available runs with different variants and labels from each of climate models, resulting in a total of 416 runs for every dust variable (120 for history and 296 for four future scenarios) and 770 runs for every meteorological variable (212 for history and 558 for four future scenarios). In addition, we collect both dust optical depth (DOD) and aerosol optical depth (AOD) at the historical periods from these models (Table S1). To facilitate the model validation and inter-comparison, we interpolate all model data with different spatial resolution to the same of 1°×1°. For each model, we average all the ensemble runs under one climatic scenario to minimize the uncertainties due to initial conditions. As a result, we derive 5 ensemble means (1 for history and 4 for future) for each variable of every model, leaving the same weight among CMIP6 models." (Lines 126-136)

**Table 1.** The information of CMIP6 models

| Model [a] | Nation | Resolution | Number of runs for dust cycle | | | | |
|---|---|---|---|---|---|---|---|
| | | | Hist | SSP126 | SSP245 | SSP370 | SSP585 |
| **CESM2-WACCM** | U.S. | 1.25°×0.94° | 3 | 1 | 5 | 3 | 5 |
| **CESM2** | U.S. | 1.25°×0.94° | 11 | 3 | 3 | 3 | 3 |
| **CNRM-ESM2-1** | France | 1.4°×1.4° | 3 | 5 | 10 | 5 | 5 |
| **GFDL-ESM4** | U.S. | 1.25°×1° | 1 | 1 | 1 | 1 | 1 |
| **GISS-E2-1-G** | U.S. | 2.5°×2° | 19 | 10 | 25 | 17 | 10 |
| **GISS-E2-1-H** | U.S. | 2.5°×2° | 10 | 5 | 5 | 1 | 5 |
| **GISS-E2-2-G** | U.S. | 2.5°×2° | 5 | 5 | 5 | 5 | 5 |
| INM-CM4-8 | Russia | 2°×1.5° | 1 | 1 | 1 | 1 | 1 |
| INM-CM5-0 | Russia | 2°×1.5° | 10 | 1 | 1 | 5 | 1 |
| MIROC-ES2L | Japan | 2.8°×2.8° | 31 | 10 | 30 | 10 | 10 |
| MIROC6 | Japan | 1.4°×1.4° | 10 | 3 | 3 | 3 | 3 |
| MRI-ESM2-0 | Japan | 1°×1° | 12 | 5 | 10 | 5 | 6 |
| **NorESM2-LM** | Norway | 2°×2° | 1 | 1 | 13 | 1 | 1 |
| **UKESM1-0-LL** | U.K. | 1.875°×1.25° | 3 | 5 | 5 | 3 | 4 |
| **Total runs** | | | 120 | 56 | 117 | 63 | 60 |

[a] The models selected for future projections are bolded.

*It is unclear what information content the regional budget column in Table 3 has. Again as everywhere, ranges should be shown for the results in this table.*

Response: The budget is calculated as the difference between emissions and depositions. The positive budget indicates a dust source while the negative budget means a dust sink. We can see that Africa is a strong source region because the emissions are higher than deposition, and the remaining dust particles have to be transported elsewhere. In contrast, all the oceans are sinks with negative budgets because they have no emissions and have to receive dust transported from land. We have revised Table 3 to show the inter-model ranges for each budget.

*Page 7, line 189: the sites offshore East Asia would certainly not be impacted from Middle Eastern dust sources*

Response: In the revised paper, we removed this explanation.

*Page 9 line 280 – "column concentration" should rather be named "column load"*

Response: We have revised accordingly.

*Some more discussion of the reasons of future precipitation and surface winds would be good.*

Response: We have explained the possible cause of increased precipitation in East Asia: "Studies have projected that global warming tends to enhance East Asian summer monsoon and South Asian summer monsoon, leading to increased precipitation in the middle and low latitudes of Asia (Sabade et al., 2011; Wang et al., 2018; Wu et al., 2022)." (Lines 340-343) We also explained the possible cause of global changes in rainfall: "We found that the main features of increased drought and wind speed over North Africa and South Africa while enhanced rainfall over Asia was retained, following the 'drier in dry and wetter in wet' pattern due to the land-air interactions through water and energy exchange (Feng and Zhang, 2015)." (Lines 423-427)

Reference

Aryal, Y. N. and Evans, S.: Global Dust Variability Explained by Drought Sensitivity in CMIP6 Models, J. Geophys. Res.: Earth Surf., 126, e2021JF006073, doi: 10.1029/2021JF006073, 2021.

Feng, H. and Zhang, M.: Global land moisture trends: drier in dry and wetter in wet over land, Sci Rep, 5, 18018, doi: 10.1038/srep18018, 2015.

Kok, J. F., Storelvmo, T., Karydis, V. A., Adebiyi, A. A., Mahowald, N. M., Evan, A. T., He, C., and Leung, D. M.: Mineral dust aerosol impacts on global climate and climate change, Nature Reviews Earth & Environment, 4, 71-86, doi: 10.1038/s43017-022-00379-5, 2023.

Sabade, S. S., Kulkarni, A., and Kripalani, R. H.: Projected changes in South Asian summer monsoon by multi-model global warming experiments, Theor. Appl. Climatol., 103, 543-565, doi: 10.1007/s00704-010-0296-5, 2011.

Wang, S., Wang, J., Zhou, Z., and Shang, K.: Regional characteristics of three kinds of dust storm events in China, Atmos. Environ., 39, 509-520, doi: 10.1016/j.atmosenv.2004.09.033, 2005.

Wang, T., Miao, J., Sun, J., and Fu, Y.: Intensified East Asian summer monsoon and associated precipitation mode shift under the 1.5 °C global warming target, Adv. Clim. Change Res., 9, 102-111, doi: 10.1016/j.accre.2017.12.002, 2018.

Woodward, S., Sellar, A. A., Tang, Y., Stringer, M., Yool, A., Robertson, E., and Wiltshire, A.: The simulation of mineral dust in the United Kingdom Earth System Model UKESM1, Atmos. Chem. Phys., 22, 14503-14528, doi: 10.5194/acp-22-14503-2022, 2022.

Wu, C., Lin, Z., and Liu, X.: The global dust cycle and uncertainty in CMIP5 (Coupled Model Intercomparison Project phase 5) models, Atmos. Chem. Phys., 20, 10401-10425, doi:

10.5194/acp-20-10401-2020, 2020.

Wu, Q., Li, Q., Ding, Y., Shen, X., Zhao, M., and Zhu, Y.: Asian summer monsoon responses to the change of land–sea thermodynamic contrast in a warming climate: CMIP6 projections, Adv. Clim. Change Res., 13, 205-217, doi: 10.1016/j.accre.2022.01.001, 2022.

Zhao, A., Ryder, C. L., and Wilcox, L. J.: How well do the CMIP6 models simulate dust aerosols?, Atmos. Chem. Phys., 22, 2095-2119, doi: 10.5194/acp-22-2095-2022, 2022.

---

## Author Comment (AC2)

**Response to the review comment 2 on acp-2022-760**

*RC2: 'Comments on acp-2022-760', Anonymous Referee #2, 22 Mar 2023*

*There is a growing concern about future dust change induced by global climate change and human activity. The study be Zhao et al. has presented the future changes in global dust cycles based on the five CMIP6 models. Ten models are first used for model evaluation, and five of these models with better performance are selected for the projection. They also investigate the change in surface wind and precipitation/relative humidity, the factors associated with the dust changes in the future. The conclusions can provide a good reference to the relevant community. Most of the manuscript is well written and clearly presented. I have some comments for the authors to consider. In particular, if possible, please provide more information on the uncertainty in the model simulation of dust cycle and discuss whether the changes are significantly large in the future.*

Response: Thank you very much for your valuable comments and constructive suggestions to further improve our manuscript. We have carefully considered all the comments and revised our manuscript accordingly. Our responses to each comment are summarized below.

*Major comments:*

*Line 366 (solid): The change of future vegetation change due to both climate change and human activity is not considered in this study, which may induce large uncertainty in the projection of future dust change. I suggest the vegetation change should be considered as well. If the impacts of vegetation change are not included, the authors should add some discussions on this.*

Response: Thank you for your suggestions. Inclusion of dynamic vegetation is expected to be an advance for the CMIP6 models. At the starting point, we hope to include vegetation cover changes as one of the drivers for future dust cycle. However, the changes of dust source areas are not provided by CMIP6 archives. In addition, the dynamic vegetation processes may instead introduce more uncertainties for the dust projections. As a result, we made great efforts to validate the simulated dust variables (concentrations and AOD, Figures 1-3) and selected the best models for future projections. In the introduction section, we have clarified as follows: "Compared to CMIP5 models, more dust emission schemes are coupled with dynamic vegetation in the CMIP phase 6 (CMIP6) models to optimize land surface emission processes (Zhao et al., 2022). Such improvement may instead amplify the uncertainties of dust simulations, because the predicted vegetation change may be inconsistent with the observed tendencies (Wu et al., 2020). As a result, it is important to validate the simulated present-day dust cycle before the application of different models in the future projection (Aryal and Evans, 2021)." (Lines 100-106)

We added Table S5 to compare the changes in dust emissions on the same vegetation-free grids as suggested: "Previous studies have revealed that dynamic vegetation process could significantly alter future dust activity (Woodward et al., 2022). However, we were not able to identify such effects because CMIP6 models do not output the information of dust sources and their strength. As a check, we compared the changes of dust emissions at vegetation-free grid points for both historical and future periods so as to exclude the impacts of vegetation changes. We found very limited differences for those grids (Table S5) relative to the changes for all grids (Table 4), suggesting that the changes of dust area are limited in most of the CMIP6 models." (Lines 387-395)

**Table S5.** Multi-model ensemble projection of the absolute (Tg a$^{-1}$) and relative changes (%) in dust emissions by the end of this century (2090-2099) at vegetation-free grid points

| Region | SSP1-2.6 | | SSP2-4.5 | | SSP3-7.0 | | SSP5-8.5 | |
|--------|----------|----------|----------|----------|----------|----------|----------|----------|
| | Absolute | Relative | Absolute | Relative | Absolute | Relative | Absolute | Relative |
| NAF | 10.1±121.7 | 1.2 | 5.8±131.5 | 0.7 | 4.2±174.1 | 0.5 | 47.4±178.9 | 5.6 |
| TGD | -0.4±23.5 | -0.8 | -2.5±41.3 | -4.9 | -6.2±53.6 | -11.9 | -4.6±55.7 | -8.9 |
| MEWA | -0.7±43.1 | -0.3 | -4.5±66.4 | -1.8 | -4.4±81.0 | -1.8 | 6.8±87.2 | 2.7 |
| AUS | 1.2±16.9 | 2.9 | 2.0±20.5 | 5.1 | -0.1±47.2 | -0.3 | 4.3±51.6 | 10.7 |
| NAM | 0.03±4.7 | 2.2 | 0.02±6.1 | 1.3 | 0.01±5.4 | 0.8 | 0.02±5.7 | 1.4 |
| SAM | 0.01±32.3 | 0.2 | 0.4±42.1 | 6.7 | -0.1±31.2 | -2.1 | -0.4±27.7 | -6.2 |
| SAF | 0.2±4.1 | 2.3 | 0.5±4.1 | 6.1 | 0.7±10.6 | 9.0 | 0.9±5.0 | 11.4 |

\* The domain of each region is shown in Fig. 4a

*Introduction and Conclusions and discussion: Some studies on dust cycle using CMIP6 models should be included for discussion:*

*Checa-Garcia et al. (2021, https://acp.copernicus.org/articles/21/10295/2021/),*

*Le and Bae (2022, https://acp.copernicus.org/articles/22/5253/2022/),*

*Li and Wang (2022, https://acp.copernicus.org/articles/22/7843/2022/),*

*Maki et al. (2022, https://www.jstage.jst.go.jp/article/sola/18/0/18_2022-035/_article/-char/ja/),*

*Woodward et al. (2022, https://acp.copernicus.org/articles/22/14503/2022/).*

Response: Thank you for your suggestions. We have included the above references in different parts of our study:
"The recent phase 6 of CMIP (CMIP6) includes more complete dust variables (e.g., emissions, depositions, concentrations, and optical depth) from climate models. The

ensemble of CMIP6 simulations has been used to depict historical changes in dust cycle and explore the possible climatic drivers (Le and Bae, 2022; Li and Wang, 2022)." (Lines 95-98)

"The predicted annual dust emissions of 2566±1996 Tg is close to the estimate of 2836 Tg yr$^{-1}$ using an ensemble of five different dust models (Checa-Garcia et al., 2021)." (Lines 401-403)

"Previous studies have revealed that dynamic vegetation process could significantly alter future dust activity (Woodward et al., 2022). … As a check, we compared the changes of dust emissions at vegetation-free grid points for both historical and future periods so as to exclude the impacts of vegetation changes." (Lines 387-392)

*Selection of models for future projection: UKESM1-0-LL may produce too much dust emission compared to other models, according to Figure S7. I am wondering if it is reasonable to select UKESM1-0-LL.*

Response: Yes, the UKESM1-0-LL produces too much dust emissions. However, this model shows reasonable performance in simulating both dust concentrations and AOD as revealed in Figure 1. As a result, we could not exclude it artificially. For this study, we use the ensemble median approach so that the extreme values from a single model will not affect the main conclusions. We have clarified in the manuscript as follows: "We applied the multi-model ensemble approach to minimize the projection biases from individual models. We used the median instead of mean values from the selected models so that our projections reflected the tendency of the majority models rather than that of the single model with maximum changes." (Lines 396-399)

*Uncertainty: As Table 3, the values of the range should be also provided for understanding the uncertainty. In addition, please provide the values for each models in supplemental files to compare different models.*

Response: We have added the inter-model range in the revised Tables 3 and 4. We also showed values for each model in Table S3.

**Table 3.** The summary of dust cycle at present day[*]

| Region | Emission | Dry Deposition | Wet Deposition | Budget[**] |
|---|---|---|---|---|
| | Tg a$^{-1}$ | Tg a$^{-1}$ | Tg a$^{-1}$ | Tg a$^{-1}$ |
| Africa | 1713±1288 | 1091±1235 | 236±155 | 386±87 |
| Asia | 736±458 | 432±419 | 226±161 | 77±32 |
| Australia | 165±237 | 110±211 | 20±25 | 35±13 |

| | | | | |
|---|---|---|---|---|
| South America | 52±106 | 30±63 | 21±23 | 1±30 |
| North America | 15±27 | 13±31 | 9±20 | -6±25 |
| Europe | 5±3 | 12±4 | 34±15 | -41±19 |
| Pacific Ocean | / | 14±12 | 48±23 | -62±33 |
| Indian Ocean | / | 46±23 | 71±36 | -117±47 |
| Atlantic Ocean | / | 95±39 | 155±57 | -250±62 |
| Arctic Ocean | / | 0±0.3 | 2±1 | -3±1 |

[*] Values from individual climate models are shown in Table S3

[**] Budget = Emission - Dry Deposition - Wet Deposition

**Table 4.** Multi-model ensemble projection of the absolute (Tg a$^{-1}$) and relative changes (%) in dust emissions by the end of this century (2090-2099)

| Region | SSP1-2.6 | | SSP2-4.5 | | SSP3-7.0 | | SSP5-8.5 | |
|---|---|---|---|---|---|---|---|---|
| | Absolute | Relative | Absolute | Relative | Absolute | Relative | Absolute | Relative |
| NAF | 10.1±121.7 | 1.2 | 5.3±131.4 | 0.6 | 4.8±148.0 | 0.6 | 47.4±178.8 | 5.6 |
| TGD | -0.4±23.5 | -0.8 | -2.5±41.3 | -4.9 | -6.2±53.6 | -11.9 | -4.6±55.7 | -8.9 |
| MEWA | -0.7±43.1 | -0.3 | -4.5±66.4 | -1.8 | -4.4±81.1 | -1.8 | 6.8±87.2 | 2.7 |
| AUS | 1.1±17.0 | 2.8 | 2.1±20.7 | 5.1 | -0.1±47.2 | -0.4 | 4.3±51.6 | 10.7 |
| NAM | 0.03±4.7 | 2.2 | 0.02±6.1 | 1.3 | 0.01±5.4 | 0.8 | 0.02±5.7 | 1.4 |
| SAM | 0.02±32.3 | 0.3 | 0.4±42.1 | 6.7 | -0.1±31.3 | -2.0 | -0.4±27.7 | -6.1 |
| SAF | 0.2±4.1 | 2.1 | 0.5±4.2 | 5.5 | 0.9±11.4 | 9.9 | 0.9±5.0 | 10.3 |

* The domain of each region is shown in Figure 1a

*Line 259: significant: How to determine the regions with significant changes?*

Response: We revised the sentence to clarify: "We select four main source regions where dust emissions are projected to increase by at least 1 Tg a$^{-1}$ under most of future climatic scenarios (Table 4). In these regions, we quantify the sensitivity of dust emissions to perturbations in meteorological factors (Fig. 7)." (Lines 293-296)

**\*\*\*\*\*\*\*\*\*\*\*\*\*\*\*\***

*Specific comments:*

*Line 22: meteorological conditions: meteorological conditions can affect the vegetation cover, which further affects the dust emission. But the impacts of vegetation change on dust emission are not mentioned in the study. Please clarify.*

Response: Please check our responses to your major comment 1.

*Line 33: relative humidity: I think soil moisture is the variable more closely related to dust emission.*

Response: Yes. We have removed relative humidity in the revised manuscript.

*Line 35 (central Asia and Taklimakan): The regions are not correctly named. According to Figure 3a, central Asia and Taklimakan should be East Asia (at least Gobi Deserts are not located in central Asia); Middle East should be Middle East and central Asia.*

Response: Following your comment, we changed the original "central Asia and Taklimakan" to "Taklimakan and Gobi Desert" and the original "Middle East" to "Middle East and West Asia".

*Line 39: due to: I think it is "partly due to".*

Response: Corrected it as suggested.

*Line 40: "As a result" should be "In total"?*

Response: Corrected it as suggested.

*Lines 65-67: First, according to Munktsetseg et al. (2016), it is more precise to say "soil moisture". Second, soil moisture alone does not control threshold friction velocity and dust emission intensity. Many factors including soil moisture determines them.*

Response: We revised the sentence as follows: "Atmospheric humidity has a tight coupling effect with soil moisture, which in part controls the threshold of friction velocity and dust emission intensity (Munkhtsetseg et al., 2016)." (Lines 65-67)

We also found that relative humidity was less important than precipitation and wind speed for dust emissions. In the revised paper, we removed the relationships between dust emissions and relative humidity in Figure 7 and clarified as follows: "Specifically, almost all the 10 region labels with reduced dust emissions under the four scenarios show increased regional precipitation but decreased wind speed, though 8 labels show decreased relative humidity (Fig. 6). It suggests that changes in precipitation and wind speed play more dominant roles in the changes of dust emissions." (Lines 288-292)

[Figure]

**Figure 7.** Relationships between the changes of dust emissions and the changes of meteorological factors. Each column represents a source region, including North Africa (NAF), Taklimakan and Gobi Deserts (TGD), Middle East and West Asia (MEWA), and Australia (AUS). Each row represents a meteorological factor, including surface wind (top) and precipitation (bottom).

*Line 114: All: it may be better to mention the date when the data are accessed to, as more data may come out later.*

Response: We added the date as suggested: "We select all available CMIP6 models (last access: April 20th, 2023) providing complete variables" (Lines 119-120)

*Lines 161-163: It is not clear to me. Please check.*

Response: We revised the sentence as follows: "In CNRM-ESM2-1, $f_m$ and $\alpha$ are combined to calculate $U_{*t}$ rather than acting as individual factors in the emission function" (Lines 175-177)

*Lines 176-177: It is hard for me to check in Fig. 2b. Perhaps also provide a table with these values in supplemental file.*

Response: We added Table S2 to list all the correlation coefficients and normalized standard deviations of individual models.

**Table S2.** The normalized standard deviations and correlation coefficients for individual models shown in Figure1

| | Dust concentrations | | AOD | |
|---|---|---|---|---|
| Model | Normalized standard deviations | Correlation coefficient | Normalized standard deviations | Correlation coefficient |
| CESM2-WACCM | 0.78 | 0.86 | 0.87 | 0.64 |
| CESM2 | 0.74 | 0.87 | 0.89 | 0.6 |
| CNRM-ESM2-1 | 0.44 | 0.85 | 0.28 | 0.79 |
| GFDL-ESM4 | 0.54 | 0.81 | 0.59 | 0.58 |
| GISS-E2-1-G | 0.76 | 0.83 | 0.55 | 0.67 |
| GISS-E2-1-H | 0.62 | 0.83 | 0.51 | 0.66 |
| GISS-E2-2-G | 1.34 | 0.84 | 0.95 | 0.61 |
| INM-CM4-8 | 0.11 | 0.49 | 0.42 | 0.44 |
| INM-CM5-0 | 0.09 | 0.30 | 0.38 | 0.26 |
| MIROC-ES2L | 0.24 | 0.82 | 0.64 | 0.3 |
| MIROC6 | 0.07 | 0.86 | 0.44 | 0.64 |
| MRI-ESM2-0 | 2.16 | 0.86 | 0.82 | 0.33 |
| NorESM2-LM | 0.33 | 0.82 | 0.68 | 0.63 |
| UKESM1-0-LL | 1.03 | 0.88 | 0.36 | 0.75 |

*Line 194: Figure 3 captions: Please also mention the latitudes and longitudes for the three regions.*

Response: We added the latitudes and longitudes of each box in Figure 4 caption: "The box regions on (a) are dust sources of North Africa (NAF) (15°N-33°N, 15°W-35°E), Middle East and West Asia (MEWA) (17°N-48°N, 40°E-70°E), Taklimakan and Gobi Deserts (TGD) (37°N-47°N, 77°E-112°E), Australia (AUS) (33°S-21°S, 113°E-144°E), North America (NAM) (28°N-37°N, 120°W-109°W), South America (SAM) (50°N-20°N, 74°S-60°S), and South Africa (SAF) (34°S-18°S, 14°E-26°E)."

*Line 230: dominates: It is not clear to me. And it is hard for me to read this from Figure 4c.*

Response: It has been revised as follows: "Furthermore, dust emissions over Asia (including Taklimakan, Gobi Deserts, West Asia and Middle East) decrease in most scenarios especially for SSP3-7.0, in which the regional reduction causes the global decline of dust emissions (Fig. 5c). " (Lines 259-261)

*Lines 232-233: But dust emission may be sensitive to precipitation change. Please clarify.*

Response: We revised the sentence to clarify: "For North Africa, regional precipitation shows mild reductions under all four scenarios even though the baseline rainfall is very low." (Lines 265-266)

*Lines 242-244: 18 regions: please check whether the numbers are correct.*

Response: It's not 18 regions but 18 region labels. We clarified as follows: "Among the total of 18 region labels (the red labels on Fig. 6) with increased dust emissions under the four scenarios, 14 labels show decreased relative humidity by at least 0.5%, 14 labels show decreased precipitation, and 10 labels show increased wind speed." (Lines 275-278)

*Line 252: limited changes: It is not clear to me.*

Response: We have removed the descriptions of relative humidity and clarified as follows: "For Middle East and West Asia, the slight increase of precipitation (Fig. 6) overweighs the moderate increase of surface wind speed, leading to a decline of regional dust emissions for SSP1-2.6 and SSP2-4.5 (Fig. 6)." (Lines 286-288)

*Lines 323-325: The sentence does not read clearly. Please revise*

Response: We revised the sentence to clarify: "The changes of dust loading in general follow that of emissions but with joint impacts of precipitation, which affects the loading through wet deposition. The decreased precipitation may further promote dust loading over regions with increased emissions (e.g. South Africa) through the reductions in wet deposition. In contrast, increased precipitation decreases dust loading by more wet deposition over regions with moderate or limited changes in dust emissions (e.g., East Asia). " (Lines 355-361)

*Lines 579-580: Not exactly red/blue (but light red & blue). I think the colors are too light to distinguish easily.*

Response: We have darkened the color of each bar shown on Figure 6.

*Figures 6 and 8: Could you make the zero lines bolder? It is not easy to see.*

Response: Corrected as suggested.

*Table 2: u_*t and u_t are different. Please clarify.*

Response: We have changed $U_t$ to $U_{*t}$. $U_{*t}$ is the threshold values.

Reference

Aryal, Y. N. and Evans, S.: Global Dust Variability Explained by Drought Sensitivity in CMIP6 Models, J. Geophys. Res.: Earth Surf., 126, e2021JF006073, doi: 10.1029/2021JF006073, 2021.

Checa-Garcia, R., Balkanski, Y., Albani, S., Bergman, T., Carslaw, K., Cozic, A., Dearden, C., Marticorena, B., Michou, M., van Noije, T., Nabat, P., O'Connor, F. M., Olivié, D., Prospero, J. M., Le Sager, P., Schulz, M., and Scott, C.: Evaluation of natural aerosols in CRESCENDO

Earth system models (ESMs): mineral dust, Atmos. Chem. Phys., 21, 10295-10335, doi: 10.5194/acp-21-10295-2021, 2021.

Le, T. and Bae, D.-H.: Causal influences of El Niño–Southern Oscillation on global dust activities, Atmos. Chem. Phys., 22, 5253-5263, doi: 10.5194/acp-22-5253-2022, 2022.

Li, W. and Wang, Y.: Reduced surface fine dust under droughts over the southeastern United States during summertime: observations and CMIP6 model simulations, Atmos. Chem. Phys., 22, 7843-7859, doi: 10.5194/acp-22-7843-2022, 2022.

Munkhtsetseg, E., Shinoda, M., Gillies, J. A., Kimura, R., King, J., and Nikolich, G.: Relationships between soil moisture and dust emissions in a bare sandy soil of Mongolia, Particuology, 28, 131-137, doi: 10.1016/j.partic.2016.03.001, 2016.

Woodward, S., Sellar, A. A., Tang, Y., Stringer, M., Yool, A., Robertson, E., and Wiltshire, A.: The simulation of mineral dust in the United Kingdom Earth System Model UKESM1, Atmos. Chem. Phys., 22, 14503-14528, doi: 10.5194/acp-22-14503-2022, 2022.

Wu, C., Lin, Z., and Liu, X.: The global dust cycle and uncertainty in CMIP5 (Coupled Model Intercomparison Project phase 5) models, Atmos. Chem. Phys., 20, 10401-10425, doi: 10.5194/acp-20-10401-2020, 2020.

Zhao, A., Ryder, C. L., and Wilcox, L. J.: How well do the CMIP6 models simulate dust aerosols?, Atmos. Chem. Phys., 22, 2095-2119, doi: 10.5194/acp-22-2095-2022, 2022.